# Artemis: Structured Visual Reasoning for Perception Policy Learning

**Wei Tang** [1 2]   **Yanpeng Sun** [3]   **Shan Zhang** [4 5]   **Weihao Bo** [1]   **Xiaofan Li** [6]   **Piotr Koniusz** [7 5]   **Wei Li** [8]   **Na Zhao** [3]   **Zechao Li** [1]

## Abstract

Recent reinforcement-learning frameworks for visual perception policy usually incorporate intermediate reasoning chains expressed in natural language. Empirical observations indicate that such purely linguistic intermediate reasoning often reduces performance on perception tasks. We argue that the core issue lies not in reasoning per se but in the form of reasoning: while these chains perform semantic reasoning in an unstructured linguistic space, **visual perception requires reasoning in a spatial and object-centric space**. In response, we introduce **Artemis**, a perception-policy learning method that performs structured visual reasoning, where each intermediate step is represented as a (label, bounding-box) pair capturing a verifiable visual state. This design enables explicit tracking of intermediate states, direct supervision for proposal quality, and avoids ambiguity introduced by language-based reasoning. Building upon verifiable and spatially grounded reasoning chains, Artemis provides a unified architecture for diverse perceptual tasks, without requiring the task-specific designs relied upon by prior perceptual policy models. Trained using grounding and detection sampeles in natural image domains, Artemis generalizes to counting and geometric perception tasks. At its core, a spatially grounded, object-centric chain rule provides a principled foundation for scalable and general perceptual policies. The source code are available at https://github.com/WayneTomas/Artemis.

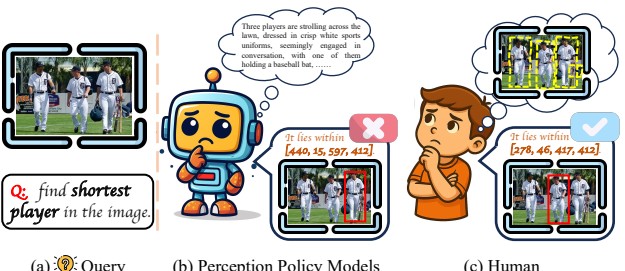

*Figure 1.* Motivation of **Artemis**. Comparison between current perception-policy models and human perception. (a) Query: find the shortest player. (b) Perception–policy models depend on ungrounded language reasoning, leading to wrong localization. (c) Humans perform structured visual reasoning, progressively refining attention to identify the correct player.

## 1. Introduction

Large language models (LLMs) (Dubey et al., 2024; Team et al., 2024b; OpenAI, 2023) have recently entered a new stage of reasoning development. Rule-based reinforcement learning (RL) (OpenAI, 2023; Team et al., 2023; 2024a) demonstrates that explicit, verifiable rewards can effectively shape reasoning behavior in large language models. Models such as DeepSeek-R1 (Guo et al., 2025a) and OpenAI-o1 (Jaech et al., 2024) provide concrete evidence that reinforcement learning can improve reasoning beyond supervised learning. Most existing progress along this line has been confined to the language modality (Team et al., 2024b; OpenAI, 2022; Team et al., 2023; 2024a; Zhu et al., 2024).

Encouraged by these advances, recent studies begin to apply rule-based RL in multimodal large language models (MLLMs) (Sun et al., 2024; Bai et al., 2025a; Wu et al., 2024; Tang et al., 2026b) during post-training to improve visual perception. Early works (Huang et al., 2025; Wu et al., 2024; Wang et al., 2024) followed the same design as reasoning-oriented LLMs, inducing models to explicitly think through language during training. Although this strategy brings moderate gains on reasoning-like tasks, it performs poorly on most perception benchmarks (Shen et al., 2025; Liu et al., 2025a; Bai et al., 2025b). As shown in Fig. 1 (b), the perception-policy model generates verbose textual thoughts unrelated to the query, leading to incorrect spatial grounding. The underlying cause is that the thinking process remains confined to a linguistic search space, disconnected

[1]Department of Computer Science and Engineering, Nanjing University of Science and Technology [2]Department of Information and Artificial Intelligence, Yangzhou University [3]IMPL, Singapore University of Technology and Design [4]AIML, The University of Adelaide [5]Data61♥CSIRO [6]Zhejiang University [7]The University of New South Wales [8]SenseTime Research. Correspondence to: Yanpeng Sun <yanpengsun115@gmail.com>.

*Proceedings of the 43rd International Conference on Machine Learning*, Seoul, South Korea. PMLR 306, 2026. Copyright 2026 by the author(s).

from spatial and object-centric representations.

To address this limitation, recent studies explore no-think (Yu et al., 2025a; Zhan et al., 2025) and adaptive-think (Li et al., 2025; Guo et al., 2025b) strategies that reduce or skip explicit reasoning during reinforcement learning. Both aim to mitigate the drawbacks of language-centric thinking and have shown improvements on several perception-related benchmarks. Although effective within specific tasks, current implementations are designed for isolated perception settings (Yu et al., 2025a; Shen et al., 2025; Zhan et al., 2025; Huang et al., 2025; Liu et al., 2025a;b). Such task-specific designs mask the limited generalization of prior models, while a general perception system should operate consistently across diverse perceptual tasks (from grounding to counting), and across heterogeneous visual domains (e.g. both natural images and diagrams). It also reveals a deeper issue in perceptual policy learning: which factors fundamentally constrain perceptual model generalization and necessitate task-tailored implementations? Instead of removing thinking altogether, we must ask: *what form of thinking truly benefits perception?*

Inspired by cognitive science (Torralba et al., 2006; Wolfe & Horowitz, 2017), we rethink how humans perceive complex scenes. As shown in Fig. 1(c), humans perceive complex scenes through visual analysis that scans the scene, locates relevant regions, and refines attention to targets before answering. Unlike linguistic reasoning, which operates in semantic space, this process grounds reasoning in spatial and object-centric representations. We refer to it as **structured visual reasoning**. Toward this end, we propose **Artemis**, a method that applies rule-based RL in MLLM post-training for *structured visual reasoning* in perception-policy learning. Specifically, we implement structured visual reasoning by representing thinking step as a set of (label–bounding-box) pairs that explicitly capture visual evidence. This explicit formulation grounds the thinking process in spatial space, enabling direct supervision on both intermediate reasoning and final answer.

To effectively learn structured visual reasoning, Artemis builds upon the strong semantic understanding already encoded in the base MLLM and focuses on teaching the model how to *see* and *ground* what it can already comprehend. Rather than optimizing linguistic reasoning, the policy learns to translate semantic concepts into object-centric visual evidence. To ground this reasoning in spatial representations, the model must learn from tasks that explicitly couple language with perception. We therefore adopt visual grounding, which requires both linguistic understanding and precise spatial localization, and has been widely recognized as one of the most comprehensive tasks for evaluating multimodal perception and reasoning. We further incorporate object detection (Tang et al., 2025; Zhan et al., 2025; Kamath

et al., 2021; Tang et al., 2026a) to provide dense scene-level supervision that strengthens the model's perceptual foundation. Both tasks are optimized under a unified rule-based reinforcement framework built upon group relative policy optimization (GRPO) (Guo et al., 2025a). We redesign the rewards to evaluate not only final outcomes but also intermediate visual reasoning steps. We validate Artemis through extensive experiments across 7 perception and 10 general multimodal benchmarks. Experimental results show that Artemis generalizes robustly across perception tasks and visual domains, adapting from grounding to counting and from natural images to mathematical diagrams. Remarkably, it also improves performance on general MLLM benchmarks, indicating that stronger perception enhances overall multimodal capability. Unlike prior approaches that require a dedicated model for each perception task, a single Artemis policy performs consistently across all settings. These findings highlight a paradigm shift: *perception policies benefit most not from eliminating thinking, but from structuring it spatially*.

## 2. Related Works

**RL-based Post-training for Reasoning.** Reinforcement learning (RL) becomes a key post-training strategy for enhancing reasoning ability in LLMs (Jaech et al., 2024; OpenAI, 2023; Team et al., 2023; 2024a). Reinforcement learning from human feedback (RLHF) uses human preference data to optimize the model policy (Ouyang et al., 2022), while direct preference optimization (DPO) (Rafailov et al., 2023; Team et al., 2024b; Dubey et al., 2024) and proximal policy optimization (PPO) (Schulman et al., 2017; Cai et al., 2024) update model policies based on preference or reward models. More recently, reinforcement learning with verifiable rewards (RLVR) attracts significant attention, with group relative policy optimization (GRPO) (Guo et al., 2025a; Shao et al., 2024) as a representative method that uses relative rewards among sampled reasoning trajectories without an explicit reward model to directly and verifiably supervise reasoning. These advances inspire similar efforts in MLLMs, such as the Qwen series (Team et al., 2024b; Bai et al., 2025a) and DeepSeek-VL2 (Wu et al., 2024). However, these approaches mainly target task-specific visual reasoning, e.g., visual math reasoning (Huang et al., 2025; Chen et al., 2025) or document understanding (Xiong et al., 2025; Yu et al., 2025b), with limited focus on perception-driven reasoning (Yu et al., 2025a).

**Visual Perception in Multimodal Models.** Visual perception has long been studied in computer vision (Sun et al., 2025b;a; Cong et al., 2022; Tu et al., 2024; Wang et al., 2026; Zhu et al., 2023), where models learn to recognize objects and reason about their spatial layout and relations (Tang et al., 2024; Kamath et al., 2021). Recently, MLLMs achieve notable progress in visual perception (Bai et al.,

2025a; Li et al., 2024; Liu et al., 2024a; Cai et al., 2024). To further improve perception, recent works apply RL for perception-policy learning, transferring reasoning-oriented strategies from LLMs and encouraging models to think explicitly in natural language. Early perception-policy works (Liu et al., 2025b; Huang et al., 2025; Bai et al., 2025b; Shen et al., 2025; Zhang et al., 2025a; Yang et al., 2025) follow this approach, improving both reasoning and perception abilities in task-specific settings, but showing limited gains on general perception tasks due to their linguistic search space. More recent methods (Li et al., 2025; Guo et al., 2025b; Zhan et al., 2025; Yu et al., 2025a) reduce or skip explicit linguistic reasoning during RL to mitigate language-centric limitations, yet improvements remain limited and largely task-specific. A related line of grounded reasoning works introduces visual evidence through region revisiting, visual re-encoding, or large-scale grounded data construction (Su et al., 2025; Sarch et al., 2025; Zheng et al., 2025; Deitke et al., 2025a), but still lacks direct supervision over intermediate object-centric reasoning states, leaving open how perceptual reasoning should be represented and supervised. From the perspective of cognitive science on how humans perceive and reason about complex scenes (Torralba et al., 2006; Wolfe & Horowitz, 2017), we rethink perception-policy learning with RL to learn a unified perception-policy that generalizes across visual perception tasks.

# 3. Artemis method

## 3.1. Preliminaries

Reinforcement learning with RLVR has recently gained attention as a post-training method for LLMs. Unlike PPO, which relies on a reward model, GRPO eliminates the need for a separate evaluation model and uses rule-based rewards to directly and verifiably guide policy optimization. Let $\mathcal{O}$ denote the model's output. For each input, the model samples a group of $G$ outputs $\{o_i\}_{i=1}^{G}$, where $o_i \in \mathcal{O}$. Each output $o_i$ receives a scalar reward $r_i$. In GRPO, the reward for each output $o_i$ is compared against the mean reward of all outputs in the group. The advantage $A_i$ is calculated as:

$$A_i = \frac{r_i - \text{mean}(r_1, \ldots, r_G)}{\text{std}(r_1, \ldots, r_G)}. \quad (1)$$

The objective function for GRPO is then:

$$J_{\text{GRPO}}(\theta) = \mathbb{E}\Big[ \min\Big(\tfrac{\pi_\theta}{\pi_{\theta_{\text{old}}}} A_i,\ \text{clip}\big(\tfrac{\pi_\theta}{\pi_{\theta_{\text{old}}}}, 1-\epsilon, 1+\epsilon\big) A_i\Big) \\ - \beta\ \text{KL}[\pi_\theta \| \pi_{\text{ref}}]\Big]. \quad (2)$$

Here, $\pi_\theta$ denotes the current policy, $\pi_{\theta_{\text{old}}}$ denotes the policy before the update, $\epsilon$ controls the clipping range for stable optimization, $\beta$ weights the KL regularization, and $\pi_{\text{ref}}$ is the reference policy. For brevity, we omit the dependence of the policies on the sampled output $o_i$.

GRPO leverages rule-based rewards that are directly verifiable, making it well-suited for visual perception tasks with explicit evaluation rules (e.g., grounding, detection). This allows effective post-training with minimal data, outperforming standard supervised fine-tuning (SFT) in both efficiency and generalization (Shen et al., 2025; Yu et al., 2025a; Li et al., 2025).

## 3.2. Rethinking Rewards for Perception Policies

Rewards are the foundation of perception-policy reinforcement learning (Shen et al., 2025; Zhan et al., 2025; Liu et al., 2025a; Yu et al., 2025a). They define the learning objective and influence how the model learns to reason. In this context, the reward is not a performance metric but a learning signal that guides how the policy is optimized.

In most current perception-policy reinforcement learning setups, the reward targets only the final outcome (Yu et al., 2025a; Zhan et al., 2025; Li et al., 2025). Such outcome-only reward design provides sparse and delayed feedback, giving no guidance on whether the model gathers or verifies the right visual evidence. Consequently, models tend to produce superficially plausible linguistic rationales instead of engaging in spatially grounded reasoning (Shen et al., 2025; Liu et al., 2025a).

This reveals a deeper issue: the misalignment between the reward objective and the perceptual process limits generalization. When the task demands spatial reasoning but the reward evaluates only semantic correctness, the learning signal encourages the wrong behavior. Evidence from process supervision studies in language models (e.g., OpenAI PRM (Lightman et al., 2023), DeepSeek-R1 (Guo et al., 2025a)) supports this view: supervising intermediate steps rather than final answers yields more robust reasoning behaviors. Conversely, forcing language-based "thinking" in perception tasks without grounding often degrades performance because the reward reinforces linguistic fluency rather than perceptual correctness.

These findings indicate that the design of reward functions (Shen et al., 2025; Yu et al., 2025a; Zhan et al., 2025) directly shapes the learning trajectories of perception policies. As most base models are pretrained in semantic reasoning spaces (Bai et al., 2025a; Wu et al., 2024; Team et al., 2024a; 2023; OpenAI, 2022; Cai et al., 2024), outcome-only rewards leave exploration unconstrained and tend to reinforce semantic priors, producing reasoning that appears coherent in language but remains visually ungrounded. In contrast, object-centric intermediate rewards balance exploration and exploitation, stabilize learning, and encourage perceptual reasoning that generalizes across tasks.

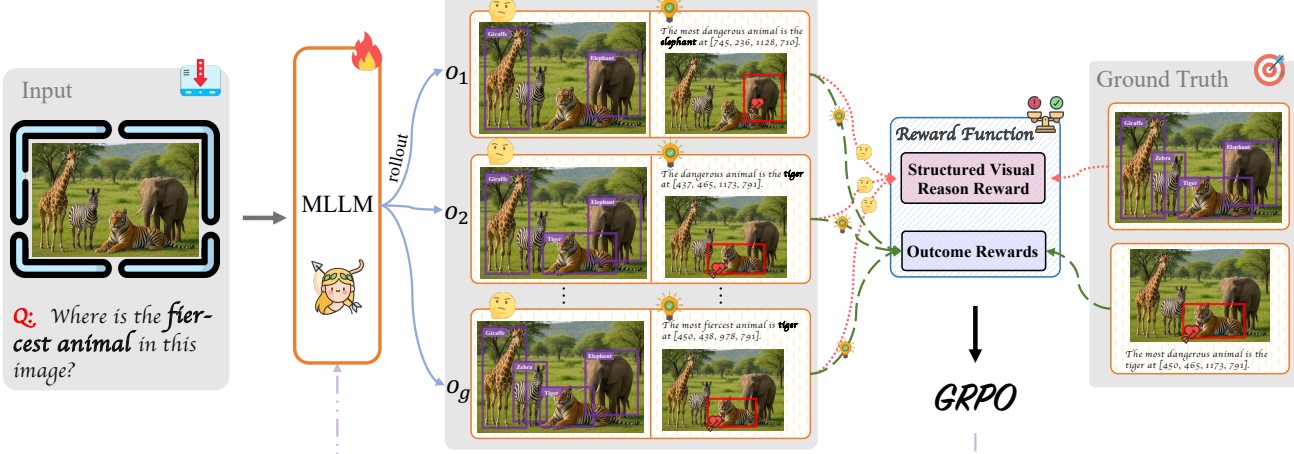

*Figure 2.* Overview of the Artemis model for RL-based perception-policy learning. Rollouts generated by MLLM are encouraged to perceive structured visual evidence before decision-making, guided by the structured visual reasoning reward, while the outcome rewards supervise the format and answer generation. GRPO is employed to optimize the unified perception-policy learning framework.

### 3.3. Artemis Reward Design

Based on the previous analysis, improving perception requires a reward that guides both the reasoning process and the final outcome. We therefore develop a unified reward framework that evaluates not only final predictions but also intermediate object-centric reasoning outputs (Fig. 2). Each rollout contains structured (label, bounding-box) pairs representing visual evidence supervised by the **Structured Visual Reasoning Reward**, while other components such as the final answer or output format are optimized with the **outcome rewards**.

**Structured Visual Reasoning Reward.** The Structured Visual Reasoning reward provides direct supervision to the model's reasoning process. It encourages explicit reasoning in the spatial domain by requiring the model to identify relevant objects and regions before producing the final answer. This object-centric formulation ensures that each reasoning step corresponds to verifiable entities in the scene rather than ambiguous linguistic descriptions. Formally, the model is prompted to output structured visual evidence within the `<think>` block:

$$\texttt{<think>...</think>} \rightarrow \left[ \{ \hat{\mathcal{C}}, \hat{\mathcal{B}} \} \right].$$

Here, $(\hat{\mathcal{C}}, \hat{\mathcal{B}})$ denote the predicted label and bounding boxes in the `<think> </think>` output.

Correspondingly, the ground-truth annotations adopt the same structured format, where each reasoning object is represented as a (`label`, `bbox`) pair. For instance, the ground-truth annotations in Fig. 2, where each purple bounding box is associated with its corresponding label:

```
"reasoning_objects":
 {{"label": "tiger", "bbox": [450, 467, 1173, 791]},
  {"label": "elephant", "bbox": [745, 236, 1128, 710]},
  {"label": "zebra", "bbox": [265, 314, 493, 768]},
  {"label": "giraffe", "bbox": [45, 33, 360, 782]}}
```

To encourage the model to generate informative structured evidence within think block, we introduce the concept of a *key object*, which corresponds to the target object in the answer (i.e. the tiger in Fig. 2). Predicted boxes that match this key object are assigned the highest reward, as correctly identifying it provides direct support for producing a correct final answer. At the same time, other predicted objects (i.e., the elephant, the zebra, and the giraffe in Fig. 2) are also assigned positive, smaller weights, ensuring that the model perceives contextual information that may be spatially or semantically related to the key object. In practice, objects whose positions and categories are similar to the key object are more likely to be relevant to the answer, so including these additional evidence objects encourage the model to produce richer, context-aware reasoning without overwhelming the supervision signal.

Formally, the weight assigned to each ground-truth box is defined as

$$w(\mathcal{B}_j) = \begin{cases} \eta, & \text{if } \mathcal{B}_j \text{ is the key object,} \\ \dfrac{1-\eta}{N-1}, & \text{otherwise,} \end{cases} \quad (3)$$

where $N$ denotes the total number of annotated objects in the scene, and $\eta$ denotes the constant score of the key object.

Since the structured visual reasoning output contains multiple predicted objects, we need to establish one-to-one correspondences with the ground-truth boxes to compute the reward accurately. Inspired by previous work involving multi-object matching (Yu et al., 2025a; Zhan et al., 2025), this is performed using the Hungarian algorithm (Kuhn, 1955), aligning each predicted box $\hat{\mathcal{B}}$ with a ground-truth box $\mathcal{B}$ and each predicted label $\hat{\mathcal{C}}$ with the corresponding ground-truth label $\mathcal{C}$.

The structured visual reasoning reward is then defined as a

weighted sum over the matched pairs:

$$r_{\text{rsn}} = \sum w(\mathcal{B}) \, \mathbf{1} \Big[ \text{IoU}(\hat{\mathcal{B}}, \mathcal{B}) \geq \tau_{\text{IoU}} \Big] \, \mathbf{1} \Big[ \text{sim}(\hat{\mathcal{C}}, \mathcal{C}) \geq \tau_{\text{sim}} \Big],$$
(4)

where $\tau_{\text{IoU}}$ and $\tau_{\text{sim}}$ denote the IoU and ROUGE similarity thresholds, respectively, and are applied consistently across all experiments. The reward is bounded in $[0, 1]$ by the weight design in Eq. (3), as GRPO relies on relative advantages rather than absolute reward values, and bounding the reward provides stable and sufficient supervision without requiring additional numerical shaping (Yu et al., 2025a; Zhan et al., 2025). During training, the model tends to generate a large number of candidate visual evidence; however, low-quality evidence can destabilize the reinforcement learning reward process. To address this, we apply high thresholds on both IoU and ROUGE to select high-quality evidence as informative contexts, thereby stabilizing RL training.

**Outcome Rewards.** To guide the model toward correct and structured outputs, following prior work (Shen et al., 2025; Yu et al., 2025a; Li et al., 2025), we leverage outcome rewards composed of two components: the format reward $r_{\text{format}}$ and the answer reward $r_{\text{ans}}$. The format reward $r_{\text{format}}$ enforces structural validity of the outputs. It checks whether the model produces a complete `<think></think>` block for intermediate reasoning (when applicable) and a valid `<answer></answer>` block for the final prediction. This reward is task-agnostic and applies to any setting that adopts structured visual reasoning. The answer reward $r_{\text{ans}}$ measures the perceptual accuracy of the model's final prediction. We use object detection to provide dense spatial supervision over the scene, while visual grounding ensures that intermediate reasoning aligns with the target object. The reward jointly evaluates localization accuracy using GIoU and label consistency to ensure correct object identification. Implementation details for both components are provided in **Appendix C**.

**Overall Reward.** The total reward combines the outcome rewards with our structured visual reasoning component:

$$r_{\text{total}} = r_{\text{format}} + r_{\text{ans}} + r_{\text{rsn}}$$
(5)

This addition encourages high-quality intermediate object-centric reasoning while maintaining accurate and well-formatted final predictions. For clarity, each component is bounded within $[0, 1]$, following the common practice in previous GRPO-based perception policy works (Shen et al., 2025; Zhan et al., 2025; Li et al., 2025; Bai et al., 2025b; Liu et al., 2025a). Notably, we find uniform reward weighting yields stable convergence and competitive performance. Consequently, no exhaustive hyper parameter tuning is performed.

# 4. Experiments

## 4.1. Implementation Details

**Training Dataset.** To supervise structured visual reasoning in our unified Artemis model, we construct the **Artemis-RFT** dataset from MS-COCO (Lin et al., 2014), resulting in roughly 77k post-training instances. Grounding and detection data are roughly balanced. For the grounding data, we build upon the answers from RefCOCO/+/g series and additionally provide ground-truth annotations specifically for computing structured visual reasoning derived from existing RefCOCO and COCO annotations. Sample examples are shown in Section 3.3. For the detection data, we adopt the COCO detection annotations with minor modifications. We also use 80k COCO detection instances as cold-start data. These do not overlap with Artemis-RFT. The detailed construction process and statistics of Artemis-RFT are provided in **Appendix D**.

**Evaluation Datasets.** To verify improvements in perception, we test our models on diverse downstream tasks and benchmarks. In-domain benchmarks include RefCOCO/+/g for visual grounding (Tang et al., 2024; Kamath et al., 2021) and COCO2017 val for object detection (Lin et al., 2014; Ren et al., 2016); zero-shot out-of-domain tasks and benchmarks include Lisa-grounding (Lai et al., 2024), visual counting (Pixmo-Count (Deitke et al., 2025b)), mathematical diagrams perception (MATHGLANCE (Sun et al., 2026)), and general ability benchmarks of MLLMs from VLMEvalKit (Duan et al., 2024) such as MMBench (Liu et al., 2023), MMVet (Yu et al., 2023), SEEDBench (Li et al., 2023), OCRBench (Liu et al., 2024b), etc.

**Model and Training Settings.** We use Qwen2.5-VL-3B-Instruct as our baseline. Training is conducted on $4 \times$ NVIDIA A100 80GB GPUs. To help the model adapt to the prompt format at early stages, and improve performance, we first perform a 1-epoch cold-start on the 80k COCO detection instances. We then train 1 epoch on the Artemis -RFT dataset. Other GRPO optimization settings follow VLM-R1 (Shen et al., 2025): rollout $G = 8$, max response length $= 2048$, temperature $= 1.0$, KL coefficient $\beta = 0.04$, initial learning rate $= 1\text{e}-6$, and global batch size $= 128$. Prompt details are provided in **Appendix E**. Empirically, we set the thresholds as $\eta = 0.8$, $\tau_{\text{IoU}} = 0.8$, and $\tau_{\text{sim}} = 0.9$, which are sufficient to demonstrate our motivation and achieve strong performance, guiding the model to both precise localization and accurate reasoning.

## 4.2. Main Results

In this subsection, we present the main experimental results and analysis; additional results, in-depth analyses, and visualizations are provided in **Appendix B and F**.

**Results of Visual Grounding.** Table 1 reports top-1 ac-

*Table 1.* Visual grounding accuracy on the RefCOCO/+/g datasets, including three IoU threshold of accuracy metrics (e.g., @50 indicates a threshold of 0.5). Models marked with † denote results from our own inference, and those highlighted in gray indicate expert models.

| Method | Size | RefCOCO | | | | | | | | | | | |
|---|---|---|---|---|---|---|---|---|---|---|---|---|---|
| | | val@50 | testA@50 | testB@50 | val@75 | testA@75 | testB@75 | val@95 | testA@95 | testB@95 | val_Avg | testA_Avg | testB_Avg |
| *Expert Models* | | | | | | | | | | | | | |
| MDETR (Kamath et al., 2021) | - | 87.5 | 90.4 | 82.6 | - | - | - | - | - | - | - | - | - |
| OFA (Wang et al., 2022) | - | 88.4 | 90.6 | 83.3 | - | - | - | - | - | - | - | - | - |
| *General MLLMs* | | | | | | | | | | | | | |
| LLaVA-v1.5 (Liu et al., 2024a) | 7B | 49.1 | 54.9 | 43.3 | 10.7 | 13.6 | 6.9 | 0.4 | 0.3 | 0.3 | 20.1 | 22.9 | 16.8 |
| LLaVA-OV (Li et al., 2024) | 7B | 73.0 | 82.3 | 63.5 | 24.2 | 29.6 | 15.9 | 0.5 | 0.5 | 0.5 | 32.6 | 37.5 | 26.6 |
| Qwen2-VL (Team et al., 2024b) | 2B | 86.8 | 89.6 | 82.0 | 77.2 | 80.6 | 70.1 | 33.0 | 35.7 | 26.9 | 65.7 | 68.6 | 59.7 |
| Qwen2.5-VL† (Bai et al., 2025a) | 3B | 88.6 | 91.7 | 84.0 | 79.1 | 83.5 | 71.2 | 34.6 | 37.9 | 27.8 | 67.4 | 71.0 | 61.0 |
| DeepSeek-VL2-Tiny† (Wu et al., 2024) | 3B | 83.5 | 86.7 | 77.9 | 69.7 | 74.1 | 60.0 | 24.6 | 29.2 | 19.3 | 59.3 | 63.3 | 52.4 |
| *RL-based MLLMs* | | | | | | | | | | | | | |
| Perception-R1 (Yu et al., 2025a) | 2B | 89.1 | 91.4 | 84.5 | 79.5 | 83.6 | 72.4 | 35.0 | 38.5 | 28.8 | 67.9 | 71.2 | 61.9 |
| Vision-R1† (Zhan et al., 2025) | 7B | 89.6 | 92.9 | 84.9 | 80.0 | 84.7 | 72.6 | 33.6 | 36.8 | 28.6 | 67.7 | 71.5 | 62.0 |
| VLM-R1† (Shen et al., 2025) | 3B | 90.7 | 92.8 | 85.9 | 81.6 | 84.7 | 73.5 | 35.6 | 37.9 | 27.7 | 69.3 | 71.8 | 62.4 |
| **Artemis** | 3B | **91.3** | **93.4** | **87.0** | **83.6** | **86.4** | **76.5** | **40.1** | **42.8** | **33.4** | **71.7** | **74.2** | **65.6** |

| Method | Size | RefCOCO+ | | | | | | | | | | | |
|---|---|---|---|---|---|---|---|---|---|---|---|---|---|
| | | val@50 | testA@50 | testB@50 | val@75 | testA@75 | testB@75 | val@95 | testA@95 | testB@95 | val_Avg | testA_Avg | testB_Avg |
| *Expert Models* | | | | | | | | | | | | | |
| MDETR (Kamath et al., 2021) | - | 81.1 | 85.5 | 72.9 | - | - | - | - | - | - | - | - | - |
| OFA (Wang et al., 2022) | - | 81.3 | 87.1 | 74.2 | - | - | - | - | - | - | - | - | - |
| *General MLLMs* | | | | | | | | | | | | | |
| LLaVA-v1.5 (Liu et al., 2024a) | 7B | 42.4 | 49.7 | 36.4 | 9.8 | 12.4 | 6.4 | 0.5 | 0.5 | 0.2 | 17.6 | 20.8 | 14.3 |
| LLaVA-OV (Li et al., 2024) | 7B | 65.8 | 79.0 | 57.2 | 23.6 | 28.8 | 15.3 | 0.6 | 0.6 | 0.4 | 30.0 | 36.1 | 24.3 |
| Qwen2-VL (Wang et al., 2024) | 2B | 77.1 | 82.5 | 70.1 | 68.7 | 73.8 | 60.0 | 29.4 | 32.3 | 23.0 | 58.4 | 62.9 | 51.0 |
| Qwen2.5-VL† (Bai et al., 2025a) | 3B | 81.9 | 87.3 | 74.7 | 73.2 | 79.3 | 63.9 | 32.3 | 35.8 | 25.4 | 62.5 | 67.5 | 54.7 |
| DeepSeek-VL2-Tiny† (Wu et al., 2024) | 3B | 73.3 | 81.3 | 63.5 | 61.9 | 70.2 | 49.4 | 22.1 | 27.3 | 16.1 | 52.4 | 59.6 | 43.0 |
| *RL-based MLLMs* | | | | | | | | | | | | | |
| Perception-R1 (Yu et al., 2025a) | 2B | 81.7 | 86.8 | 74.3 | 73.6 | 79.3 | 64.2 | 32.6 | 36.9 | 26.7 | 62.6 | 67.7 | 55.1 |
| Vision-R1† (Zhan et al., 2025) | 7B | 83.0 | 89.0 | 75.3 | 74.7 | 81.7 | 64.1 | 31.5 | 35.2 | 25.6 | 63.1 | 68.6 | 55.0 |
| VLM-R1† (Shen et al., 2025) | 3B | 84.2 | 89.3 | 76.6 | 76.1 | 81.2 | 65.7 | 33.4 | 36.4 | 25.9 | 64.6 | 69.0 | 56.1 |
| **Artemis** | 3B | **85.3** | **89.9** | **77.8** | **78.3** | **82.9** | **68.7** | **38.3** | **41.7** | **30.0** | **67.3** | **71.5** | **58.7** |

| Method | Size | RefCOCOg | | | | | | | |
|---|---|---|---|---|---|---|---|---|---|
| | | val@50 | test@50 | val@75 | test@75 | val@95 | test@95 | val_Avg | test_Avg |
| *Expert Models* | | | | | | | | | |
| MDETR (Kamath et al., 2021) | - | 83.3 | 83.3 | - | - | - | - | - | - |
| OFA (Wang et al., 2022) | - | 82.2 | 82.3 | - | - | - | - | - | - |
| *General MLLMs* | | | | | | | | | |
| LLaVA-v1.5 (Liu et al., 2024a) | 7B | 43.2 | 45.1 | 8.5 | 9.3 | 0.3 | 0.3 | 17.3 | 18.2 |
| LLaVA-OV (Li et al., 2024) | 7B | 70.8 | 70.8 | 23.3 | 23.6 | 0.6 | 0.7 | 31.6 | 31.7 |
| Qwen2-VL (Wang et al., 2024) | 2B | 83.3 | 83.1 | 72.7 | 73.0 | 28.9 | 27.9 | 61.6 | 61.3 |
| Qwen2.5-VL† (Bai et al., 2025a) | 3B | 85.1 | 85.7 | 74.4 | 75.8 | 32.1 | 33.1 | 63.9 | 64.9 |
| DeepSeek-VL2-Tiny† (Wu et al., 2024) | 3B | 75.7 | 79.2 | 60.4 | 63.1 | 19.1 | 21.0 | 38.8 | 54.4 |
| *RL-based MLLMs* | | | | | | | | | |
| Perception-R1 (Yu et al., 2025a) | 2B | 85.7 | 85.4 | 75.7 | 76.0 | 32.1 | 33.1 | 64.5 | 64.8 |
| Vision-R1† (Zhan et al., 2025) | 7B | 86.4 | 86.9 | 76.4 | 77.8 | 32.4 | 33.1 | 65.1 | 65.9 |
| VLM-R1† (Shen et al., 2025) | 3B | 86.0 | 86.7 | 75.1 | 76.8 | 32.7 | 32.9 | 64.6 | 65.5 |
| **Artemis** | 3B | **87.3** | **87.3** | **77.7** | **79.4** | **36.3** | **37.9** | **67.1** | **68.2** |

curacy under different IoU thresholds on RefCOCO/+/g. Compared with the base Qwen2.5-VL-3B, Artemis consistently improves performance across all splits, highlighting enhanced perception. Compared to RL-based MLLMs, including VLM-R1, which performs reasoning in task-specific models over linguistic reasoning, as well as Perception-R1 and Vision-R1, which are also task-specific but skip reasoning by directly supervising final answers, our structured visual reasoning within a unified model yields larger gains across all splits, especially at higher IoU thresholds. For instance, on RefCOCO testB, Artemis improves over VLM-R1 by 1.1 at IoU@50, over Vision-R1 by 3.9 at IoU@75, and over Perception-R1 by 4.6 at IoU@95, demonstrating that Artemis produces highly accurate bounding boxes.

*Table 2.* Object detection evaluation on COCO2017 val. Models marked with † denote results from our own inference. Gray rows indicate expert models (from MMDetection (Chen et al., 2019)).

| Method | Size | Epoch | Object detection COCO2017 Val | | | |
|---|---|---|---|---|---|---|
| | | | mAP | AP50 | AP75 | AR100 |
| *Expert Models* | | | | | | |
| YOLOv3 (Redmon & Farhadi, 2018) | - | 273 | 27.9 | 49.2 | 28.3 | - |
| Faster-RCNN (Ren et al., 2016) | - | 12 | 35.6 | 55.7 | 37.9 | - |
| *General MLLMs* | | | | | | |
| Qwen2.5-VL† (Bai et al., 2025a) | 3B | 1 | 15.4 | 22.5 | 15.9 | 29.8 |
| Griffon (Zhan et al., 2024) | 13B | 1 | 24.8 | 40.6 | 25.1 | - |
| *RL-based MLLMs* | | | | | | |
| VLM-R1† (Shen et al., 2025) | 3B | 1 | 21.6 | 35.6 | 21.7 | 33.2 |
| Vision-R1 (Zhan et al., 2025) | 7B | 1 | 26.6 | 40.0 | 27.8 | - |
| Perception-R1 (Yu et al., 2025a) | 3B | 1 | **31.9** | 46.7 | **33.4** | 41.2 |
| **Artemis** | 3B | 1 | 31.0 | **48.0** | 31.9 | **46.6** |

**Results of Object Detection.** We report object detection results on COCO2017 val, as shown in Table 2. Object

*Table 3.* Zero-shot visual counting on Pixmo-Count and reasoning grounding on LISA test. Models marked with [†] denote results from our own inference. Models marked with [‡] follow a detect-then-count paradigm, deriving counts from predicted boxes.

| Method | Size | Visual Counting | | Reasoning Grounding |
|---|---|---|---|---|
| | | Pixmo$_{val}$ | Pixmo$_{test}$ | LISA$_{test}$ |
| *General MLLMs* | | | | |
| LLaVA-v1.5 (Liu et al., 2024a) | 7B | 33.3 | 31.0 | - |
| LLaVA-OV (Li et al., 2024) | 7B | 55.8 | 53.7 | - |
| Qwen2-VL (Wang et al., 2024) | 2B | 60.2 | 50.5 | - |
| Qwen2.5-VL[†] (Bai et al., 2025a) | 3B | 58.0 | 57.8 | 67.4 |
| *RL-based MLLMs* | | | | |
| VisionReasoner[‡] (Liu et al., 2025a) | 7B | 70.1 | 69.5 | - |
| Perception-R1 [‡] (Yu et al., 2025a) | 2B | 78.1 | 75.6 | - |
| UniVG-R1 (Bai et al., 2025b) | 7B | - | - | 59.7 |
| No-Thinking-RL (Li et al., 2025) | 2B | - | - | 61.8 |
| VLM-R1 (Shen et al., 2025) | 3B | - | - | 63.1 |
| **Artemis** | 3B | **81.4** | **76.9** | **69.8** |

detection remains one of the most demanding tasks in visual perception due to its requirement for accurate localization and comprehensive scene understanding. The evaluation follows prior MLLM-based detection methods (e.g., the Griffon series and Perception-R1 (Zhan et al., 2024; 2025; Yu et al., 2025a)), where each predicted box is assigned a default confidence score of 1 and unparsable predictions are discarded. This default is used because MLLM-based models do not produce confidence scores, ensuring fair comparison with previous work. Compared with the Qwen2.5-VL-3B baseline, Artemis achieves a substantial improvement across all metrics (e.g. mAP 31.0 vs. 15.4 and AP$_{50}$ 48.0 vs. 22.5), representing a qualitative leap in perception capability. Compared with other RL-based methods, Artemis demonstrates stronger overall scene perception and more complete detection, achieving higher AP$_{50}$ and AR$_{100}$ (48.0 and 46.6) than Perception-R1 (46.7 and 41.2). Besides, Perception-R1 is optimized more specifically for COCO detection, whereas Artemis aims at a unified policy and obtains stronger transfer across grounding, counting, and diagram perception. To the best of our knowledge, Artemis is the only unified RL-based model that surpasses an mAP of 30 within the 3B scale, while showing notable advantages in both AP$_{50}$ and AR$_{100}$.

**Zero-shot Visual Perception on Natural Scenes.** We report zero-shot visual counting accuracy on the Pixmo-Count dataset (Table 3). Even under fully zero-shot conditions, without any counting-specific training, our model achieves strong performance by internally enumerating instances during the <think> phase and directly producing numeric counts in a human-like manner (for detailed visualizations, see **Appendix F.2**). This emergent ability offers a clear and intuitive improvement in perceptual reasoning. In contrast, Perception-R1, which skips the reasoning process, and VisionReasoner, which adopts language-based reasoning, both follow a detect-then-count paradigm with post-hoc aggregation of predicted boxes. Even though these methods are trained on the Pixmo dataset, making our zero-shot setting more challenging, Artemis still outperforms them.

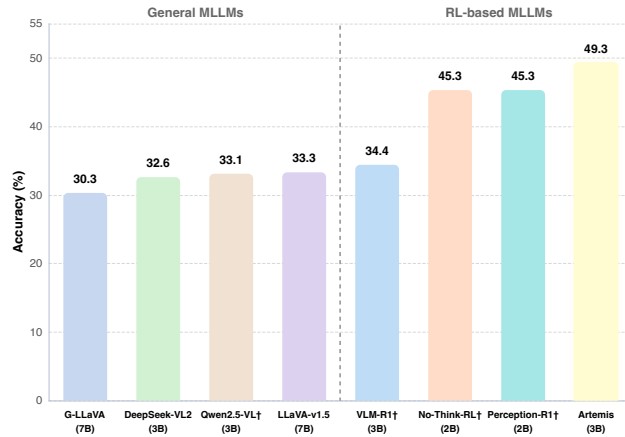

*Figure 3.* Zero-shot accuracy on MATHGLANCE benchmark. Models marked with [†] denote results from our own inference.

We further show reasoning visual grounding accuracy on the LISA test set in Table 3. Our model achieves 69.8 accuracy, substantially outperforming prior works such as UniVG-R1 (59.7), and No-Thinking-RL (61.8). This improvement demonstrates that structured visual reasoning enhances not only spatial understanding but also reasoning-dependent perception in natural scenes. This improvement demonstrates that explicitly associating spatial reasoning with the thinking process enhances grounding accuracy on contextual objects and reduces linguistic hallucinations through verifiable visual supervision, improving overall perceptual capabilities in complex natural scenes.

**Zero-shot Visual Perception on Math Scene.** Fig. 3 reports the zero-shot results on the MATHGLANCE benchmark, which evaluates visual mathematical perception across plane geometry, solid geometry, and graph-based problems. All models are evaluated using the official MATHGLANCE prompts. From these results, we can observe that Artemis achieves the best overall average score (49.3), substantially outperforming all involved general MLLMs (e.g., Qwen2.5-VL, LLaVA-v1.5) and recent R1-based models (e.g., Perception-R1, No-Thinking-RL). The improvement reflects an overall enhancement of perceptual capabilities. Importantly, the structured visual reasoning learned from natural images transfers to math-related visual scenes, demonstrating that Artemis effectively enhances perceptual capabilities with strong generalization and robustness across different types of visual tasks.

**Zero-shot General Visual Comprehension.** Table 4 reports zero-shot results on a wide range of multimodal benchmarks for general visual understanding and reasoning. After training with our structured visual reasoning, Artemis demonstrates a consistent improvement in overall performance, indicating that the model's perception ability has been strengthened and more uniformly aligned across diverse visual tasks. These results further demonstrate that our method effectively enhances the model's general visual

*Table 4.* Zero-shot comprehensive evaluation of visual perception across multiple mainstream multimodal benchmarks. Models marked with † denote results from our own inference.

| Method | Size | MMBench Avg. | MMVet Avg. | MMStar Avg. | ScienceQA Avg. | SeedBench Avg. | MME Sum | AI2D Avg. | OCRBench Avg. | POPE Avg. | BLINK Avg. |
|---|---|---|---|---|---|---|---|---|---|---|---|
| *General MLLMs* | | | | | | | | | | | |
| LLaVA-v1.5 (Liu et al., 2024a) | 7B | 62.8 | 32.8 | 32.6 | 65.4 | 60.1 | 1338.3 | 51.9 | - | - | - |
| Qwen2-VL (Wang et al., 2024) | 2B | 71.9 | 45.6 | 46.3 | 74.0 | 72.7 | 1471.1 | 71.6 | - | - | - |
| DeepSeek-VL2-Tiny (Wu et al., 2024) | 3B | 74.6 | 52.5 | - | - | - | 1905.5 | - | 805 | - | - |
| Qwen2.5-VL† (Bai et al., 2025a) | 3B | 79.1 | 60.0 | 53.8 | 79.3 | 74.0 | 2200.0 | **78.3** | 826 | 85.9 | **48.8** |
| *RL-based MLLMs* | | | | | | | | | | | |
| VLM-R1† (Shen et al., 2025) | 3B | 70.7 | 58.8 | 53.1 | 69.4 | 68.8 | 2156.2 | 73.3 | 774 | 79.3 | 46.9 |
| Perception-R1 (Yu et al., 2025a) | 2B | 71.8 | 48.9 | 45.7 | 73.4 | 73.0 | 1903.9 | 71.8 | - | - | - |
| **Artemis** | 3B | **79.3** | **61.4** | **55.9** | **79.6** | **74.3** | **2229.7** | 78.2 | **828** | **88.6** | 48.5 |

*Table 5.* Ablation study on the contributions of Grd. (Reward for Grounding), Det. (Reward for Detection), and S.V. Rsn. (Structured Visual Reasoning Reward). The last column shows the average MATHGLANCE accuracy.

| Grd. | Det. | S.V. Rsn. | RefCOCOg$_{val}$ | COCO$_{mAP}$ | MATHGLANCE |
|---|---|---|---|---|---|
| - | - | - | 85.1 | 15.4 | 33.1 |
| ✓ | | | 86.6 | 27.5 | 43.1 |
| | ✓ | | 51.6 | 30.6 | 44.0 |
| ✓ | | ✓ | 86.8 | 28.0 | 44.2 |
| ✓ | ✓ | | 87.1 | 30.7 | 43.7 |
| ✓ | ✓ | ✓ | **87.3** | **31.0** | **49.3** |

*Table 6.* Ablation study comparing different forms of reasoning on our model with cold-start initialization. "None" indicates no reasoning, "Ling. Rsn." denotes linguistic reasoning, and "S.V. Rsn." denotes structured visual reasoning.

| Forms | RefCOCOg$_{val}$ | COCO$_{mAP}$ | MATHGLANCE | Pixmo$_{val}$ | LISA$_{test}$ |
|---|---|---|---|---|---|
| None | **87.5** | 30.6 | 44.3 | 65.2 | **71.7** |
| Ling. Rsn. | 86.4 | 30.4 | 47.3 | 13.4 | 66.5 |
| S.V. Rsn. | 87.3 | **31.0** | **49.3** | **81.4** | 69.8 |

comprehension without task-specific tuning.

## 4.3. Ablation Study of Artemis

**Rewards Composition.** We conduct ablations to evaluate the contributions of different reward components on both in-domain (RefCOCOg, COCO$_{mAP}$) and out-of-domain (MATHGLANCE) perception, as shown in Table 5. Adding the grounding reward (Grd. in the table, indicates format and answer rewards for grounding task) alone improves the model's perception, particularly on out-of-domain MATH-GLANCE, indicating better generalization. Using only the detection reward (Det. in the table, indicates format and answer rewards for detection task) enhances holistic scene perception (COCO$_{mAP}$) but substantially reduces grounding accuracy. Incorporating structured visual reasoning with grounding further improves performance across all metrics, confirming that structured intermediate reasoning enhances both in-domain and out-of-domain perception. In contrast, joint training without structured visual reasoning guidance may achieve reasonable in-domain performance, but its generalization to out-of-domain scenarios is limited. Enabling all three components achieves the best balanced performance, demonstrating the complementarity of the rewards in strengthening general visual understanding.

**Reasoning Forms.** We perform an ablation comparing

*Table 7.* Ablation study comparing different training orders for grounding (Grd.) and detection (Det.) tasks. "Det. → Grd." means training detection first, then grounding; "Grd. → Det." means training grounding first, then detection; "Uni." means training both tasks simultaneously.

| Order | RefCOCOg$_{val}$ | COCO$_{mAP}$ | MATHGLANCE | Pixmo$_{val}$ | LISA$_{test}$ |
|---|---|---|---|---|---|
| Det. → Grd. | 86.9 | 28.4 | 46.7 | 77.1 | 68.8 |
| Grd. → Det. | 87.2 | 30.9 | 42.7 | 79.2 | 67.8 |
| Uni. | **87.3** | **31.0** | **49.3** | **81.4** | **69.8** |

*None reasoning*, *linguistic reasoning*, and our default *structured visual reasoning*, keeping all other training conditions identical. For the None reasoning and linguistic reasoning settings, we adjust the prompts so that the model outputs only the final answer, or produces textual reasoning steps in addition to the final answer, while supervising only the final answer. As shown in Table 6, removing reasoning entirely yields high in-domain grounding scores but poor performance on out-of-domain benchmarks such as MATH-GLANCE, Pixmo, and LISA. This is because supervising only final answers encourages overfitting to local bounding-box objectives, leading to high training-set accuracy but limited generalization. Introducing linguistic reasoning further reduces performance across most out-of-domain perception benchmarks, especially in visual counting. This suggests that the generated linguistic explanations, while superficially plausible, do not help the perceptual process and may even interfere with it. In contrast, structured visual reasoning not only preserves competitive in-domain performance but also improves perceptual generalization. By guiding the model with object-centric rewards, it translates semantic concepts into visual evidence, leading to more stable optimization and perceptual representations that generalize across tasks.

**Effect of Joint Training on Perception.** To investigate the effect of task ordering on perceptual gains, we conduct an ablation study comparing unified training with two sequential alternatives: training detection first and then grounding (Det. → Grd.), and training grounding first and then detection (Grd. → Det.). As shown in Table 7, the sequential strategies yield competitive performance on in-domain visual grounding (RefCOCOg$_{val}$), with Grd. → Det. slightly outperforming Det. → Grd. (87.2 vs. 86.9). However, unified training consistently achieves the strongest results across all out-of-domain perception benchmarks. This in-

dicates that while sequential training can handle individual tasks effectively, unified optimizing grounding and detection allows the model to learn more generalizable perceptual representations. In particular, unified training balances the supervision signals across tasks, preventing overfitting to a single objective and promoting transfer to diverse downstream perception challenges.

## 5. Conclusion

This work rethinks perception-policy learning inspired by how human perception of complex scenes, emphasizing the importance of structured visual reasoning. Building on this insight, we present Artemis, a unified rule-based RL method that guides visual perception learning through structured visual reasoning. By supervising reasoning in a structured and verifiable manner, the model develops stronger perception and demonstrates substantial generalization across across a broad range of in-domain and out-of-domain visual perception tasks. Our findings confirm that aligning intermediate reasoning with spatial and object-centric manner empowers perception-policy learning.

**Limitations.** Artemis currently adopts a simple label–bbox format to represent intermediate visual evidence and focuses on structured visual reasoning in static 2D images. Future work may explore richer forms of structured visual reasoning representations and extend this formulation beyond 2D images, such as to video-based, or 3D visual perception.

## Acknowledgements

We would like to thank Dr. Jingdong Wang at Baidu Inc. for his valuable guidance on the conceptual development and manuscript refinement of this work. This work was supported by National Natural Science Foundation of China (Grant No. 62425603), Basic Research Program of Jiangsu Province (Grant No. BK20240011) and City-University Cooperation Funding Project (Grant No. YZ2025208).

## Impact Statement

This paper presents methodological research in machine learning that emphasizes structured visual reasoning to enhance the perception capabilities of multimodal large models. By providing a structured and verifiable learning paradigm, this work aims to facilitate the development of more robust and general-purpose multimodal models by the research community. All datasets used in this work are publicly available and do not involve data from any human, animal, or private individuals, and we do not identify any specific ethical or societal concerns beyond those commonly associated with general-purpose machine learning research.

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

## A. Relation to Grounded Reasoning Methods

We provide a conceptual comparison with recent grounded reasoning methods (Su et al., 2025; Sarch et al., 2025; Zheng et al., 2025; Deitke et al., 2025a) in Table 8. Here, Re-enc. denotes visual revisiting or re-encoding during inference, and Proc. Sup. denotes deterministic supervision over intermediate perceptual states. These methods share the high-level goal of connecting MLLM reasoning with visual evidence, but differ in how visual evidence is represented and supervised. Pixel Reasoner, ViGoRL, and DeepEyes mainly rely on visual revisiting or re-encoding during inference, such as region selection, cropping, or zooming. Molmo/PixMo improves grounded perception through large-scale perception-aligned data construction and supervised fine-tuning. In contrast, Artemis keeps the visual input fixed and uses structured label–box states as intermediate visual evidence, enabling deterministic process supervision during RL.

*Table 8.* Conceptual relation to recent grounded reasoning methods.

| Method | Paradigm | Re-enc. | Proc. Sup. | Representation |
|---|---|---|---|---|
| Pixel Reasoner | RL | ✓ | – | Region revisit |
| ViGoRL | RL | ✓ | – | Point / revisit |
| DeepEyes | RL | ✓ | – | Region revisit |
| Molmo/PixMo | SFT | – | – | Point / data |
| **Artemis** | RL | – | ✓ | Label–box evidence |

## B. Supplementary Experiments

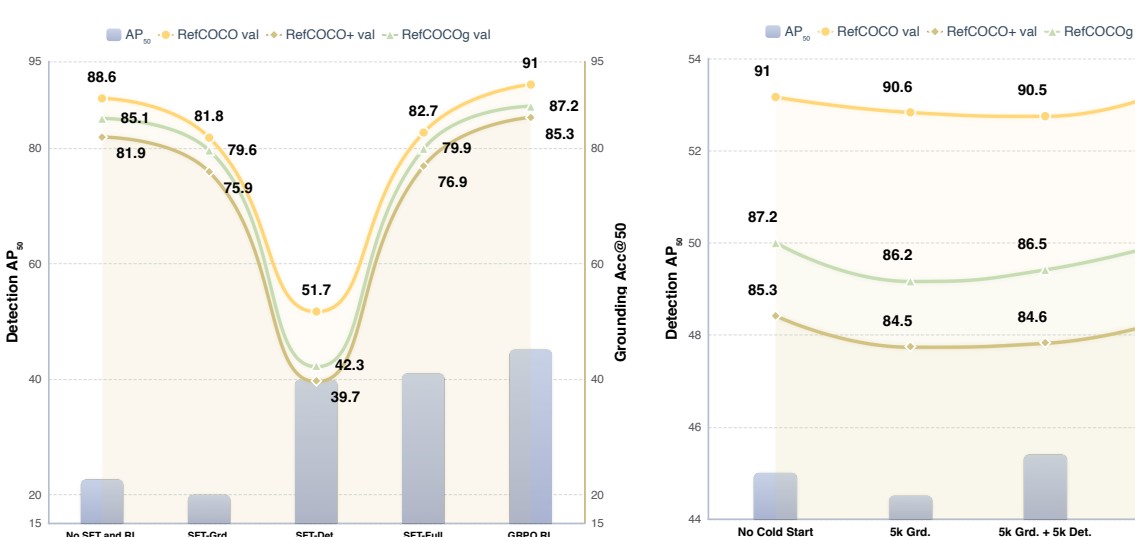

*Figure 4.* Impact of SFT and RL training strategies on COCO detection (bars) and visual grounding (lines).

*Figure 5.* Impact of cold start strategies on COCO detection (bars) and visual grounding (lines).

### B.1. Effect of SFT and RL Training Strategies

We ablate the impact of different training strategies on visual perception, as shown in Fig. 4. All models are trained on the same base model (Qwen2.5-VL-3B) and the same set of samples, with no cold-start initialization. In particular, SFT-Full and GRPO-based RL use exactly the same training samples, differing only in the output format; other SFT variants (SFT-Grd. and SFT-Det.) are trained on subsets of this data. The baseline Qwen2.5-VL-3B without SFT or RL exhibits low scene perception, especially in detection. Applying SFT only on grounding (SFT-Grd.) decreases grounding performance, likely due to insufficient data for learning structured visual reasoning. In contrast, SFT on detection (SFT-Det.) improves detection but significantly harms grounding, highlighting the challenge of balancing fine-grained perception in single-task SFT. Mixing SFT for both grounding and detection (SFT-Full) partially recovers performance, though gains remain limited. Only with GRPO-based RL can the model effectively leverage structured visual reasoning to improve both detection and grounding under the same training samples, demonstrating that reinforcement learning more effectively enhances perceptual capabilities than supervised fine-tuning.

## B.2. Cold-Start Analysis

We conduct ablation study to invest the effect of cold-start strategies and the proportion of grounding and detection data on the model's perception performance (Fig. 5). Without cold-start data, the model struggles to detect contextual objects. This is likely due to insufficient exposure to diverse scene contexts, which results in reduced scene perception. Such deficiencies may compromise downstream tasks that rely on comprehensive scene understanding. Introducing only a small amount of grounding data may provide initial guidance, but it can also disrupt the model's pre-existing capabilities learned from baseline training, and may even negatively affect other perceptual abilities, such as detection. Adding a balanced proportion of detection data significantly recovers performance, while applying cold start directly to detection achieves the best overall results. This indicates that improving scene-level perception not only enhances structured visual reasoning reasoning but also boosts the model's overall perception capabilities.

## B.3. Time Cost Comparison

We evaluate the inference efficiency of our method by comparing it with existing approaches on the RefCOCOg test split for the visual grounding task, using a batch size of 1 on a single NVIDIA A100 80GB GPU. To reduce the effect of PyTorch cold start, the first 50 samples are skipped. Then, 20 random samples are measured over 5 runs, and both the mean and standard deviation of the runtime are reported in Table 9. From the results, Perception-R1 without intermediate reasoning has the shortest runtime, but as noted in the main text, its perception performance on both in-domain and out-of-domain tasks is comparatively lower. Our method achieves similar inference time to VLM-R1 while delivering much stronger perception and reasoning capabilities. This highlights that our approach is both efficient and effective without noticeably increasing the time cost.

*Table 9.* Inference time comparison on RefCOCOg test split.

| Method | Size | Average Time (s) | Std. (s) |
|---|---|---|---|
| *RL-based MLLMs* | | | |
| Perception-R1[†] | 2B | 0.96 | 0.029 |
| VLM-R1[†] | 3B | 3.54 | 0.136 |
| **Artemis** | 3B | 4.03 | 0.185 |

## B.4. Results on Revisit-style Benchmarks

We further evaluate Artemis on revisit-style benchmarks that require fine-grained visual re-examination. As shown in Table 10, Artemis improves over Qwen2.5-VL-3B on both V* (Wu & Xie, 2024) and HR-Bench4K (Wang et al., 2025), although it does not explicitly revisit or re-encode image regions during inference. Compared with Look-back-7B (Yang et al., 2026), Artemis-3B achieves comparable relative gains over its own 3B backbone, while avoiding explicit revisit-style inference. These results suggest that structured visual reasoning can provide complementary benefits to revisit-based methods.

*Table 10.* Results on revisit-style benchmarks.

| Method | Size | V* / Time (h) | HR-Bench4K / Time (h) |
|---|---|---|---|
| Qwen2.5-VL | 7B | 77.0 / 0.2 | 68.3 / 4.0 |
| Look-back | 7B | 78.0 / 2.7 | 69.2 / 108.0 |
| Qwen2.5-VL | 3B | 75.4 / 0.1 | 66.5 / 0.9 |
| **Artemis** | 3B | 76.4 / 0.1 | 67.6 / 3.8 |

## B.5. More In-depth Analysis of Artemis

Here we provide an in-depth analysis of why Structured Visual Reasoning are fundamentally better aligned with perception-policy learning than conventional linguistic reasoning. As noted in the *Introduction* (**Main paper Section 1**), our design is grounded in the cognitive intuition of how humans perceive and reason about visual scenes (Torralba et al., 2006; Wolfe & Horowitz, 2017). Human perception operates through coordinated spatial attention, object localization, and semantic integration. Neuroscience studies show that humans process complex scenes by sequentially directing attention across spatially relevant regions (Bisley & Goldberg, 2010). The posterior parietal cortex maintains a spatial priority map that enables rapid localization of task-relevant objects before any linguistic reasoning takes place (Jerde et al., 2012). In other words, the reasoning process begins with where to look and what is present, rather than with holistic linguistic descriptions.

In our proposed Structured Visual Reasoning, the computational analogue of this "moving spotlight" is the (label–bounding-box) pair, where bounding box coordinates provide precise spatial localization and category labels indicate object identity. Linguistic reasoning, by contrast, are free-form and inherently uncertain: they tend to summarize the entire scene using broad captions that are vulnerable to hallucinated visual cues, irrelevant details (*e.g.*, color or background descriptions), and the loss of fine-grained spatial constraints. Structured Visual Reasoning instead enforce explicit, spatially grounded

reasoning that remains tightly coupled to the policy's perceptual state. Modeling the reasoning process directly in visual space and object-centric ensures that intermediate thoughts propagate in a predictable and interpretable manner, with each step anchored to a specific region of the visual input. This mismatch between linguistic reasoning and visual priority undermines the policy's stability and generalizability, further underscoring the advantage of structured visual reasoning for perception-policy learning.

We conducted comprehensive experiments to validate the effectiveness of Artemis, a unified rule-based RL method for perception–policy learning equipped with structured visual reasoning, as reported in **Section 4.2 (Main paper)**. We further provide a controlled experiment using the same training setup as ours but varying only the form of the "thinking" process, as shown in Table 6, with detailed analysis in the following section. A key finding from these evaluations concerns zero-shot generalization: structured object-centric thoughts enable the policy model to generalize to out-of-domain tasks and transition smoothly from natural images to diagrammatic inputs.

**Main paper Table 3** shows that, without any exposure to the counting task during training and without any deliberate post-hoc processing, Artemis achieves a significant improvement of +11.3 over VisionReasoner (which relies on language-based reasoning). We further provide qualitative visualizations in Fig. 11–Fig. 12. For example, in Fig. 11, our model reasons in a visual format, explicitly listing target objects and their locations, which naturally leads to the correct answer of 6. By contrast, the base Qwen2.5-VL model hallucinates multiple bounding boxes for the same balloon while missing others, yielding an incorrect count. Notably, Qwen2.5-VL is not able to directly infer the numeric quantity; instead, it relies on post-processing heuristics (`len(list)`) to obtain the final number, highlighting its lack of grounded perceptual reasoning while merely following the counting prompt to list potential targets.

**From Natural Images to Mathematical Diagrams.** Unlike natural images, mathematical diagrams are inherently semantically sparse yet structurally rich, requiring genuine understanding of geometric primitives rather than superficial pattern recognition to complete the perception tasks (Sun et al., 2026). Prior work shows that even advanced general MLLMs (*e.g.*, GPT-4o) struggle with planar geometry, especially in fine-grained grounding tasks (Sun et al., 2026). In contrast, Artemis naturally transfers its strong perceptual ability to mathematical understanding in diagrammatic settings. This zero-shot transfer improvement is non-trivial: in the tasks of MATHGLANCE benchmark, shape classification requires distinguishing highly similar polygon families (e.g., equilateral vs. right triangles), and relationship identification involves mathematical relations such as perpendicularity and parallelism that differ fundamentally from natural-image categories. Artemis's ability to perform these tasks in a zero-shot manner reinforces our claim: a stronger perceptual model should handle diverse perception tasks within a unified method, rather than relying on separately trained task-specific models for each individual task. Moreover, the Structured Visual Reasoning design provides a task-agnostic and generalizable format for perception, enabling robust transfer across domains.

## C. Details of Outcome Rewards

As described in the main paper, we teach the model structured visual reasoning by relying on tasks that explicitly couple language with perception. Visual grounding provides fine-grained supervision that links semantic concepts to precise spatial locations, while object detection offers dense scene-level signals that strengthen the model's perceptual foundation. These tasks guide the policy to translate semantic understanding into object-centric visual evidence. Under our unified rule-based reinforcement method built upon GRPO, we redesign the rewards to evaluate not only final outcomes but also intermediate visual reasoning steps. Below, we provide the details of the outcome rewards.

### C.1. Format Rewards

We define a format reward $r_{\text{format}}$ to ensure that model outputs follow the expected structure for rule-based evaluation.

**Visual Grounding.** The output must include a correctly structured `<think> </think>` block followed by an `<answer> </answer>` block containing at least one bounding box. Let $\mathcal{O}$ denote the model's output. The format reward is defined as

$$r_{\text{format}}^{\text{VG}} = \begin{cases} 1, & \text{if } \mathcal{O} \text{ is valid} \\ 0, & \text{otherwise} \end{cases} \tag{6}$$

**Object Detection.** Unlike grounding tasks, which require interpreting a user prompt to locate a specific target within the image, pure detection tasks aim to identify and localize all foreground objects based solely on their visual features, without

reasoning over candidate regions guided by language. We therefore omit the `<think> </think>` block in this setting. The format reward is

$$\mathbf{1}_{\text{valid}}(\mathcal{O}) = \begin{cases} 1, & \mathcal{O} \text{ is valid,} \\ 0, & \text{otherwise,} \end{cases} \tag{7}$$

$$r_{\text{format}}^{\text{DET}} = \begin{cases} 1, & \mathbf{1}_{\texttt{think}}(\mathcal{O}) = 0 \text{ and } \mathbf{1}_{\text{valid}}(\mathcal{O}) = 1, \\ 0.5, & \mathbf{1}_{\texttt{think}}(\mathcal{O}) = 1 \text{ and } \mathbf{1}_{\text{valid}}(\mathcal{O}) = 1, \\ 0, & \text{otherwise.} \end{cases} \tag{8}$$

where $\mathbf{1}_{\texttt{think}}(\mathcal{O})$ indicates the presence of a `<think>` block in $\mathcal{O}$, and $\mathbf{1}_{\text{valid}}(\mathcal{O})$ indicates whether $\mathcal{O}$ is valid.

Finally, we can define the overall format reward across tasks as:

$$r_{\text{format}} = r_{\text{format}}^{\text{VG}} + r_{\text{format}}^{\text{DET}}. \tag{9}$$

This unified formulation allows us to handle both tasks in a single model while respecting their individual format requirements.

### C.2. Answer Rewards

Building on the format rewards, the answer rewards directly evaluate the perceptual correctness of model predictions.

**Visual Grounding.** For visual grounding, each sample contains exactly one target region. Given the predicted bounding box $\hat{\mathcal{B}}$ and the ground-truth bounding box $\mathcal{B}$, the answer reward measures localization quality using the GIoU (Zhang et al., 2025b):

$$r_{\text{ans}}^{\text{VG}} = \text{GIoU}(\hat{\mathcal{B}}, \mathcal{B}). \tag{10}$$

**Object Detection.** Object localization often involves multiple objects, requiring one-to-one correspondences between predictions and ground-truth boxes to accurately compute rewards. Following Vision-R1 (Zhan et al., 2025), we use the Hungarian algorithm (Kuhn, 1955) to establish these correspondences between predicted bounding boxes $\hat{\mathcal{B}}$ with predicted categories $\hat{\mathcal{C}}$ and ground-truth boxes $\mathcal{B}$ with categories $\mathcal{C}$. Based on these matched pairs, the base detection reward is defined as a weighted sum of three components:

$$r_{\text{base}}^{\text{DET}} = \frac{1}{3}\Big(\text{GIoU}(\hat{\mathcal{B}}, \mathcal{B}) + \mathbf{1}\{\hat{\mathcal{C}} = \mathcal{C}\} + \text{F1}(\hat{\mathcal{B}}, \mathcal{B})\Big). \tag{11}$$

The first term measures localization quality, the second checks exact label match, and the third captures coverage completeness over all predicted and ground-truth boxes.

To improve detection quality, we apply two multiplicative penalties to the base detection reward. The *quantity penalty* encourages the number of predicted bounding boxes to match the number of ground-truth boxes:

$$\rho_{\text{num}}(\hat{\mathcal{B}}, \mathcal{B}) = \frac{\min(|\hat{\mathcal{B}}|, |\mathcal{B}|)}{\max(|\hat{\mathcal{B}}|, |\mathcal{B}|)}, \tag{12}$$

and the *missing penalty* penalizes each ground-truth box $b \in \mathcal{B}$ that is not matched by any prediction $\hat{b} \in \hat{\mathcal{B}}$:

$$\rho_{\text{miss}}(\hat{\mathcal{B}}, \mathcal{B}) = 1 - \frac{|\{b \in \mathcal{B} \mid b \text{ is unmatched by any } \hat{b} \in \hat{\mathcal{B}}\}|}{|\mathcal{B}|}. \tag{13}$$

The final detection answer reward is then computed as

$$r_{\text{ans}}^{\text{DET}} = r_{\text{base}}^{\text{DET}} \cdot \rho_{\text{num}}(\hat{\mathcal{B}}, \mathcal{B}) \cdot \rho_{\text{miss}}(\hat{\mathcal{B}}, \mathcal{B}), \tag{14}$$

and is clipped to lie within $[0, 1]$ for stability.

Similarly, we can define the overall answer reward across tasks as the sum of the task-specific rewards:

$$r_{\text{ans}} = r_{\text{ans}}^{\text{VG}} + r_{\text{ans}}^{\text{DET}}. \tag{15}$$

## D. Dataset Construction

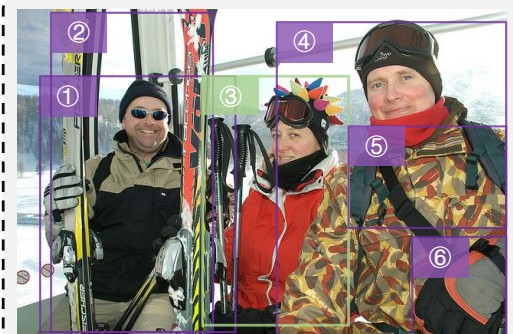

**Problem:**
a white woman skier with a colorful hat sitting between two men skiers.
**Solution:**
[245, 85, 431, 411]
**Reasoning_objects:**
[[\"a man wearing a black and beige coat and blue glasses, holding skates\", [32, 85, 290, 417]],
[\"skis\", [50, 1, 263, 415]],
[\"a white woman skier with a colorful hat sitting between two men skiers\", [245, 85, 431, 411]],
[\"a man on the right wearing a camouflage parka posing on a ski lift\", [340, 15, 643, 415]],
[\"a backpack\", [433, 139, 643, 283]],
[\"a man with an orange and black backpack over his shoulder and wearing a black, orange, and gray glove on his hand\", [515, 285, 638, 411]]]
**Task_type:**
Visual grounding

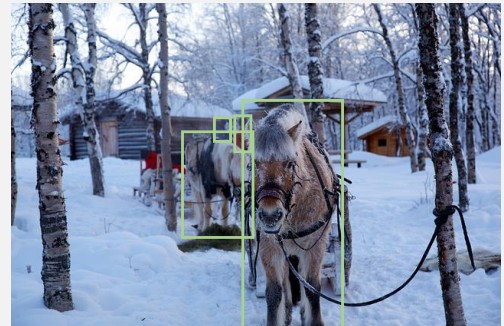

**Problem:**
List all visible objects in the image with their bounding boxes.

**Solution:**
[[\"horse\", [307, 128, 431, 419]],
[\"person\", [294, 146, 314, 191]],
[\"horse\", [229, 167, 316, 295]],
[\"person\", [266, 147, 285, 175]]]

**Task_type:**
Object detection

*Figure 6.* A data example from Artemis-RFT. Our dataset contains two task types, visual grounding and object detection, and our unified Artemis perception policy learning framework is jointly trained on both. For the grounding example, purple boxes denote the reasoning objects; the numeric labels are added only for illustration and do not exist in the raw data. The solution is indicated by a green box. In this example, box ③ is the solution and also serves as the key reasoning object, so only the green box is shown where the two overlap. For the detection example, we display only the green box that corresponds to the solution.

As described in the main paper, we supervise structured visual reasoning using a dedicated post-training dataset. To support this objective within our unified Artemis model, we construct the **Artemis-RFT** dataset from MS-COCO (Lin et al., 2014), yielding roughly 77k post-training instances. Table 11 summarizes the dataset, showing the number of samples, images, and average samples per image for the grounding and detection subsets. The grounding (Grd.) subset contains 39,651 samples from 7,037

*Table 11.* Statistics of the **Artemis-RFT** dataset, including the grounding (Grd.) and detection (Det.) subsets. We also report average samples per image.

| Partition | #Samples | #Images | Avg. Samples per Image |
|---|---|---|---|
| Grd. subset | 39,651 | 7,037 | 5.6 |
| Det. subset | 37,446 | 37,446 | 1 |
| **Total** | 77,097 | 44,483 | – |

images, averaging 5.6 samples per image, designed to teach precise spatial localization. The detection (Det.) subset contains 37,446 samples from 37,446 images, averaging 1 sample per image, providing dense scene-level supervision.

Specifically, the dataset is constructed using MS-COCO images together with grounding-related annotations from LLaVA-665k (Liu et al., 2024a) (including RefCOCO/+/g). From these annotations, we extract referring expression queries and their associated bounding boxes. Region-caption annotations associated with localized regions are also collected and used as visual evidence for constructing reasoning traces. When multiple captions describe the same region, they are merged into a single concise description. All bounding boxes are converted from normalized padded-square coordinates back to the original image resolution and unified under a consistent coordinate convention of Qwen2.5-VL (Bai et al., 2025a). To further enrich the object-centric evidence, we incorporate region annotations from Vision-R1 (Zhan et al., 2025) that correspond to the same images, standardizing their labels and coordinates to match our representation. During this process, multiple bounding boxes that refer to the same image region are merged to avoid redundant annotations while preserving all valid visual evidence. To ensure a clear correspondence between reasoning process and final answers, we make sure that every answer corresponds to a region included in the reasoning objects, which also enables the identification of key objects. For the detection subset, we randomly sample 80k COCO detection annotations as cold-start data and incorporate the remaining annotations in the same unified box format, providing additional scene-level supervision. A data example of Artemis-RFT is shown in Fig. 6.

---

**Visual Grounding (Training & Testing)**

**User**: Analyze the image and answer the following question:'What region is described as: `<description>`?'

Think step by step:

1. Identify a small set of key objects or regions that help locate the described target.

2. These may include the target object itself and/or contextual objects (e.g., nearby items, reference persons, background cues).

3. Estimate their bounding box(es) in [x1, y1, x2, y2] format (pixel coordinates).

Output your reasoning in `<think>` tags as a JSON list of key objects:

```
<think>
[{"label": "<object_label_1>", "bbox": [x1, y1, x2, y2]},
 {"label": "<object_label_2>", "bbox": [x1, y1, x2, y2]}]
</think>
```

Then output the final answer in `<answer>` tags:

```
<answer>{"label": "<description>", "bbox": [x1, y1, x2, y2]}</answer>
```

*Table 12.* Default prompt used for visual grounding in Artemis.

---

**Object Detection (Training & Testing)**

**User**: Analyze the image and answer: "List all visible objects in the image with their bounding boxes."

Think step by step:

1. Identify all major visible objects in the image from the following category set: [category str of coco].

2. For each object, estimate its category and bounding box [x1, y1, x2, y2] (pixel coordinates).

3. Avoid duplicates and irrelevant objects.

4. Directly output the final answer in `<answer>` tags as a JSON list of all detected objects, and do not ouput any `<think>`:

Directly output the final answer in `<answer>` tags:

```
<answer>
[{"label": "<object_label_1>", "bbox": [x1, y1, x2, y2]},
 {"label": "<object_label_2>", "bbox": [x1, y1, x2, y2]}]
</answer>
```

*Table 13.* Default prompt used for object detection in Artemis.

## E. Prompt Settings

For reproducibility, we provide the complete prompts used in our experiments. Table 12, Table 13, and Table 14 show the prompts for visual grounding, object detection, and visual counting, respectively.

For LISA grounding, we use the same prompt as in the visual grounding task. For MATHGLANCE and general visual comprehension MLLM evaluation benchmarks, such as MMStar, we use the official prompts provided by the evaluation kits (e.g., VLMEVALkit (Duan et al., 2024)) to ensure a fair comparison.

For the other comparison methods included in the main paper, i.e., models marked with [†], the results are generated using our own inference. We use the official checkpoints corresponding to the most suitable task (when available) and follow the prompt settings recommended in their original papers or GitHub repositories (if provided). For full details on the inference procedures, please refer to the original publications and codebases.

Visual Counting (Zero-shot Testing Only)

**User**: Analyze the image and answer: "How many '`<description>`' are in the image?"

Step 1: Examine the image carefully and list **all visible instances** of '`<description>`' in `<think>` tags.
Step 2: Each instance should be a dictionary with "label" and "bbox" if possible.
Step 3: After listing all instances, count them.
Step 4: Output the total count in `<answer>` tags. **The number must exactly equal the number of items in `<think>`. Do NOT estimate or guess.**

Example:
```
<think>
[{"label": "<description> #1", "bbox": [x1, y1, x2, y2]},
 {"label": "<description> #2", "bbox": [x1, y1, x2, y2]}]
</think>

<answer>
{"<description> count": <number_of_items_in_think>}
</answer>
```

*Table 14.* Default prompt used for zero-shot visual counting in Artemis.

**Grounding Problem: the person riding the horse that is in the center front area**

**Solution: [286.17, 58.77, 414.77, 287.02]**

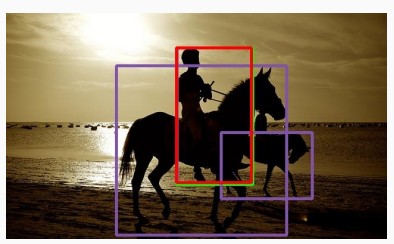 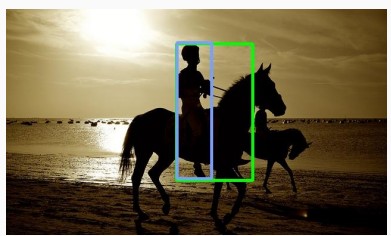 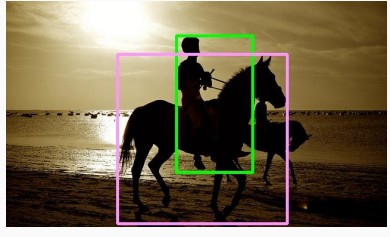

**[Artemis]**

```
<think>
[{"label": "the person riding the horse that is in the center
front area", "bbox": [289, 60, 415, 294]},
{"label": "horse", "bbox": [187, 92, 474, 385]},
{"label": "horse", "bbox": [364, 207, 518, 322]}]
</think>

<answer>
[{"label": "the person riding the horse that is in the center
front area", "bbox": [288, 60, 415, 294]}]
</answer>
```

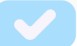

**[Perception-R1]**

the person riding the horse that is in the center front area(448,150),(539,749)

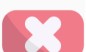

**[VLM-R1]**

```
<think>
The image shows two horses on a beach at sunset. The
primary focus is on the horse in the foreground, which
appears to be ridden by a person. This horse is closer to
the camera and is positioned slightly to the right side of the
frame.
</think>

<answer>{"bbox_2d": [189, 93, 475, 386], "label": "horse
being ridden by a person"}
</answer>
```

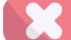

*Figure 7.* Qualitative visual grounding results (Case 1) of Artemis compared with Perception-R1 and VLM-R1 on the RefCOCO/+/g datasets. For brevity, full prompts are omitted. Green: ground truth; Purple: Artemis reasoning boxes; Red: Artemis answer; Blue: Perception-R1; Pink: VLM-R1.

# F. Qualitative Results

In this section, we provide qualitative visualizations to complement the quantitative results. We illustrate Artemis' performance on visual grounding (Section F.1), visual counting (Section F.2), and additional tasks including MATHGLANCE, COCO Val 2017, and LISA grounding (Section F.3).

## Grounding Problem: a zebra with its head on the back of another zebra

### Solution: [177.53, 29.99, 375, 208.65]

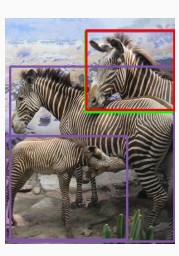 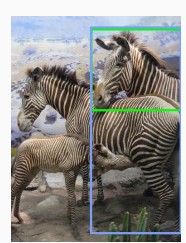 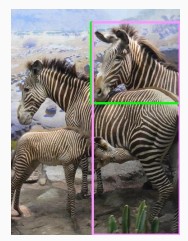

**[Artemis]**

**\<think\>**
[{"label": "a zebra with its head on the back of another zebra", "bbox": [174, 30, 363, 205]},
{"label": "zebra", "bbox": [11, 108, 363, 498]},
{"label": "zebra", "bbox": [0, 264, 262, 494]}]
**\</think\>**

**\<answer\>**
[{"label": "a zebra with its head on the back of another zebra", "bbox": [174, 28, 363, 205]}]
**\</answer\>**

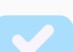

**[Perception-R1]**

a zebra with its head on the back of another zebra
(470,51),(999,960)

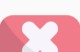

**[VLM-R1]**

**\<think\>** The image shows two zebras, one adult and one younger. The adult zebra is standing behind the younger one, with its head resting on the back of the younger zebra. This behavior is common among zebras as they seek comfort and warmth from their companions.
**\</think\>**

**\<answer\>**{"bbox_2d": [176, 30, 364, 504], "label": "a zebra with its head on the back of another zebra."}
**\</answer\>**

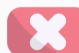

*Figure 8.* Qualitative visual grounding results (Case 2) of Artemis compared with Perception-R1 and VLM-R1 on the RefCOCO/+/g datasets. For brevity, full prompts are omitted. Green: ground truth; Purple: Artemis reasoning boxes; Red: Artemis answer; Blue: Perception-R1; Pink: VLM-R1.

## Grounding Problem: cat not image

### Solution: [173.86, 164.17, 455.18, 379.98]

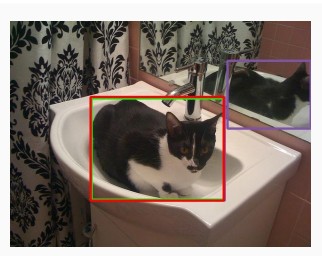 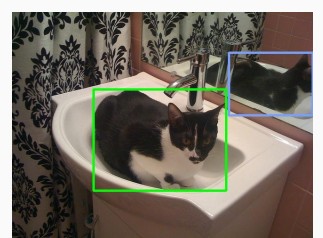 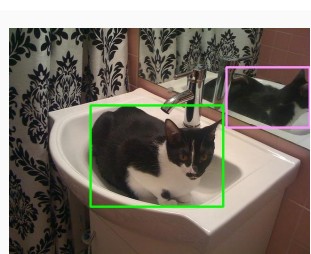

**[Artemis]**

**\<think\>**
[{"label": "cat not image", "bbox": [465, 84, 643, 227]},
{"label": "cat", "bbox": [173, 160, 459, 378]}]
**\</think\>**

**\<answer\>**
[{"label": "cat not image", "bbox": [173, 160, 457, 378]}]
**\</answer\>**

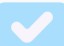

**[Perception-R1]**

the cat not image(720,180),(999,457)

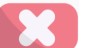

**[VLM-R1]**

**\<think\>**
The cat is sitting inside the sink, which is located in the bathroom. The sink is white and has a faucet on it. The cat appears to be looking at the camera. The bathroom also features a shower curtain with a floral pattern and a mirror above the sink.
**\</think\>**

**\<answer\>**
{"bbox_2d": [465, 83, 644, 210], "label": "cat not image."}
**\</answer\>**

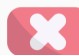

*Figure 9.* Qualitative visual grounding results (Case 3) of Artemis compared with Perception-R1 and VLM-R1 on the RefCOCO/+/g datasets. For brevity, full prompts are omitted. Green: ground truth; Purple: Artemis reasoning boxes; Red: Artemis answer; Blue: Perception-R1; Pink: VLM-R1.

## Grounding Problem: man on the frontmost bike

### Solution: [212.14, 245.16, 290.36, 466.95]

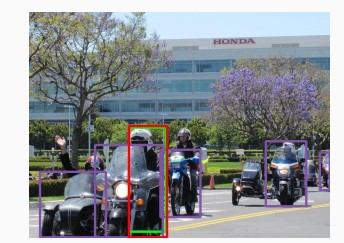 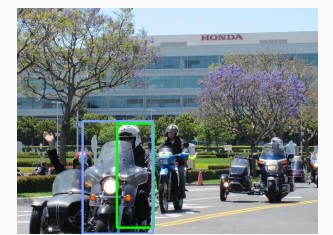 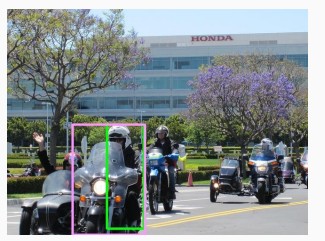

**[Artemis]**

**\<think\>**
[{"label": "man on the frontmost bike", "bbox": [214, 240, 295, 475]},
{"label": "motorcycle", "bbox": [141, 280, 286, 475]},
{"label": "motorcycle", "bbox": [281, 290, 367, 432]},
{"label": "motorcycle", "bbox": [507, 272, 593, 409]},
{"label": "motorcycle", "bbox": [24, 337, 165, 475]},
{"label": "motorcycle", "bbox": [622, 296, 643, 375]}]
**\</think\>**

**\<answer\>**
[{"label": "man on the frontmost bike", "bbox": [214, 240, 295, 475]}]
**\</answer\>**

**[Perception-R1]**

the man on the frontmost bike(217,501),(450,999)

**[VLM-R1]**

**\<think\>**
The person riding the motorcycle in the foreground is wearing a helmet and appears to be the main focus of the image. Given the context, it's likely that this individual is the one described as "man on the frontmost bike."
**\</think\>**

**\<answer\>**
{"bbox_2d": [140, 243, 295, 476], "label": "man on the frontmost bike."}
**\</answer\>**

*Figure 10.* Qualitative visual grounding results (Case 4) of Artemis compared with Perception-R1 and VLM-R1 on the RefCOCO/+/g datasets. For brevity, full prompts are omitted. Green: ground truth; Purple: Artemis reasoning boxes; Red: Artemis answer; Blue: Perception-R1; Pink: VLM-R1.

## Counting Problem: hot air balloons

### Solution: 6

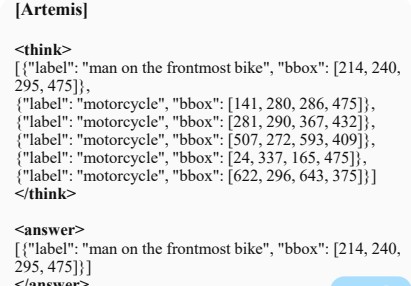 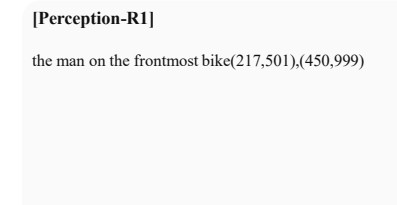 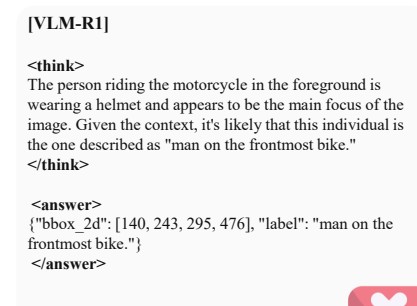

**[Artemis]**

**\<think\>**
[{\"label\": \"hot air balloons #1\", \"bbox\": [108, 79, 200, 209]},
{ \"label\": \"hot air balloons #2\", \"bbox\": [452, 361, 538, 479]},
{ \"label\": \"hot air balloons #3\", \"bbox\": [767, 78, 856, 202]},
{ \"label\": \"hot air balloons #4\", \"bbox\": [126, 659, 216, 775]},
{ \"label\": \"hot air balloons #5\", \"bbox\": [753, 637, 839, 748]},
{ \"label\": \"hot air balloons #6\", \"bbox\": [1084, 358, 1172, 476]}]
**\</think\>**

**\<answer\>**
{\"hot air balloons count\": 6}
**\</answer\>**

**[Perception-R1]**

hot air balloons
(636,96),(713,250)(83,99),(167,263)(374,446),(447,590)
(105,815),(180,951)(626,787),(696,920)(898,439),(973,589)
(0,0),(171,239)

Post-hoc count: 7

**[Qwen2.5-VL]**

```json
[
{"bbox_2d": [1085, 357, 1172, 476], "label": "hot air balloons"},
{"bbox_2d": ["129, 658", 214, 774], "label": "hot air balloons"},
{"bbox_2d": [756, 637, 836, 747], "label": "hot air balloons"},
{"bbox_2d": [768, 78, 853, 201], "label": "hot air balloons"},
{"bbox_2d": [108, 80, 200, 209], "label": "hot air balloons"},
{"bbox_2d": [129, 658, 214, 774], "label": hot air balloons"},
{"bbox_2d": [756, 637, 836, 747], "label": "hot air balloons"},
{"bbox_2d": [108, 80, 200, 209], "label": "hot air balloons},
{"bbox_2d": [768, 78, 853, 201], "label": "hot air balloons"}
]
```

Post-hoc count: 9

*Figure 11.* Qualitative visual counting results (Case 1) of Artemis compared with Perception-R1 and Qwen2.5-VL on the Pixmo datasets. For brevity, full prompts are omitted. Green: ground truth; Purple: Artemis reasoning boxes; Red: Artemis answer; Blue: Perception-R1; Cyan: Qwen2.5-VL.

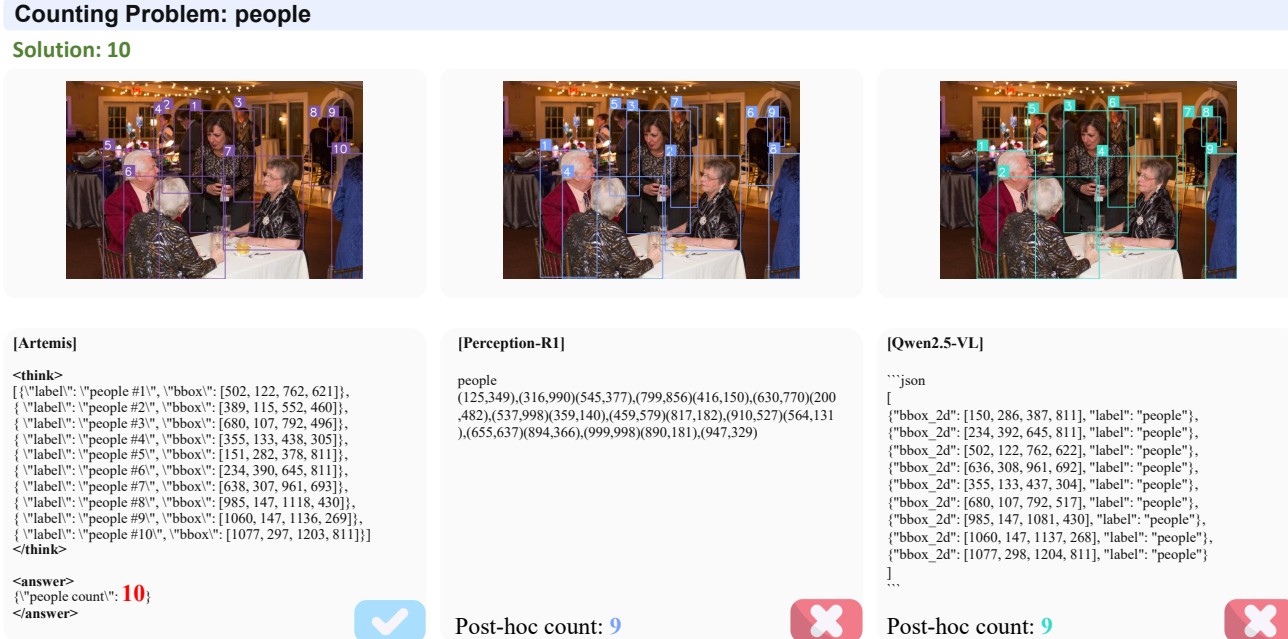

**Counting Problem: people**

**Solution: 10**

**[Artemis]**

**<think>**
[{\"label\": \"people #1\", \"bbox\": [502, 122, 762, 621]},
{ \"label\": \"people #2\", \"bbox\": [389, 115, 552, 460]},
{ \"label\": \"people #3\", \"bbox\": [680, 107, 792, 496]},
{ \"label\": \"people #4\", \"bbox\": [355, 133, 438, 305]},
{ \"label\": \"people #5\", \"bbox\": [151, 282, 378, 811]},
{ \"label\": \"people #6\", \"bbox\": [234, 390, 645, 811]},
{ \"label\": \"people #7\", \"bbox\": [638, 307, 961, 693]},
{ \"label\": \"people #8\", \"bbox\": [985, 147, 1118, 430]},
{ \"label\": \"people #9\", \"bbox\": [1060, 147, 1136, 269]},
{ \"label\": \"people #10\", \"bbox\": [1077, 297, 1203, 811]}]
**</think>**

**<answer>**
{\"people count\": **10**}
**</answer>**

**[Perception-R1]**

people
(125,349),(316,990)(545,377),(799,856)(416,150),(630,770)(200,482),(537,998)(359,140),(459,579)(817,182),(910,527)(564,131),(655,637)(894,366),(999,998)(890,181),(947,329)

Post-hoc count: **9**

**[Qwen2.5-VL]**

```json
[
{"bbox_2d": [150, 286, 387, 811], "label": "people"},
{"bbox_2d": [234, 392, 645, 811], "label": "people"},
{"bbox_2d": [502, 122, 762, 622], "label": "people"},
{"bbox_2d": [636, 308, 961, 692], "label": "people"},
{"bbox_2d": [355, 133, 437, 304], "label": "people"},
{"bbox_2d": [680, 107, 792, 517], "label": "people"},
{"bbox_2d": [985, 147, 1081, 430], "label": "people"},
{"bbox_2d": [1060, 147, 1137, 268], "label": "people"},
{"bbox_2d": [1077, 298, 1204, 811], "label": "people"}
]
```

Post-hoc count: **9**

*Figure 12.* Qualitative visual counting results (Case 2) of Artemis compared with Perception-R1 and Qwen2.5-VL on the Pixmo datasets. For brevity, full prompts are omitted. Green: ground truth; Purple: Artemis reasoning boxes; Red: Artemis answer; Blue: Perception-R1; Cyan: Qwen2.5-VL.

## F.1. Qualitative Comparisons on RefCOCO/+/g

In Fig. 7–Fig. 10, we present several visual grounding examples frome RefCOCO/+/g. For the Perception-R1 model that skips the reasoning process, its grounding performance is noticeably inaccurate across these cases. This suggests that removing the reasoning stage may hinder the model's ability to further learn and perceive complex scenes, as the model is supervised only through answer rewards. Consequently, it tends to optimize toward matching annotated boxes rather than developing general perceptual skills. As discussed in main paper, such training may yield reasonable in-domain performance but fails to substantially enhance the underlying perception capability, resulting in poor generalization to out-of-domain perception data or tasks. In essence, this approach does not fully leverage the strong reasoning potential brought by MLLMs and GRPO-based reinforcement learning, making it fundamentally similar to previous grounding models that directly optimize the final outputs.

For linguistic reasoning based VLM-R1, the main limitation lies in the difficulty of supervising perception oriented reasoning within the linguistic space, which often leads to mismatches between the reasoning and the final answer. For example, in Case 1 (Fig. 7), the query is "the person riding the horse in the center front area". Although the reasoning appears to correctly describe the image, the answer incorrectly shifts the target from the person to the horse. In Case 2 (Fig. 8), the reasoning is already incorrect at the start, stating that there are two zebras while the image actually contains three. In Case 3 (Fig. 9) and Case 4 (Fig. 10), the model provides the correct descriptions of the scenes during the reasoning stage, yet the final answer still selects an incorrect or imprecise object. These reasoning and answer mismatches, together with the quantitative results, indicate that linguistic reasoning may not provide an effective or reliable reasoning mechanism for perception driven grounding tasks.

Together with the quantitative results, the visualization cases verify that structured visual reasoning rewards provide a more suitable training signal for perception tasks. By guiding the model to reason in an object-centric manner, as shown by the purple reasoning bounding boxes and red answer bounding boxes in the cases, our Artemis produces more precise and correct grounding results compared with other methods like Perception-R1 and VLM-R1.

## F.2. Qualitative Comparisons on Visual Counting

In Fig. 11 and Fig. 12, we present zero-shot visual counting qualitative results of Artemis on the Pixmo dataset, compared with Perception-R1 (trained on Pixmo as described in the main paper) and the baseline Qwen2.5-VL-3B. As discussed in

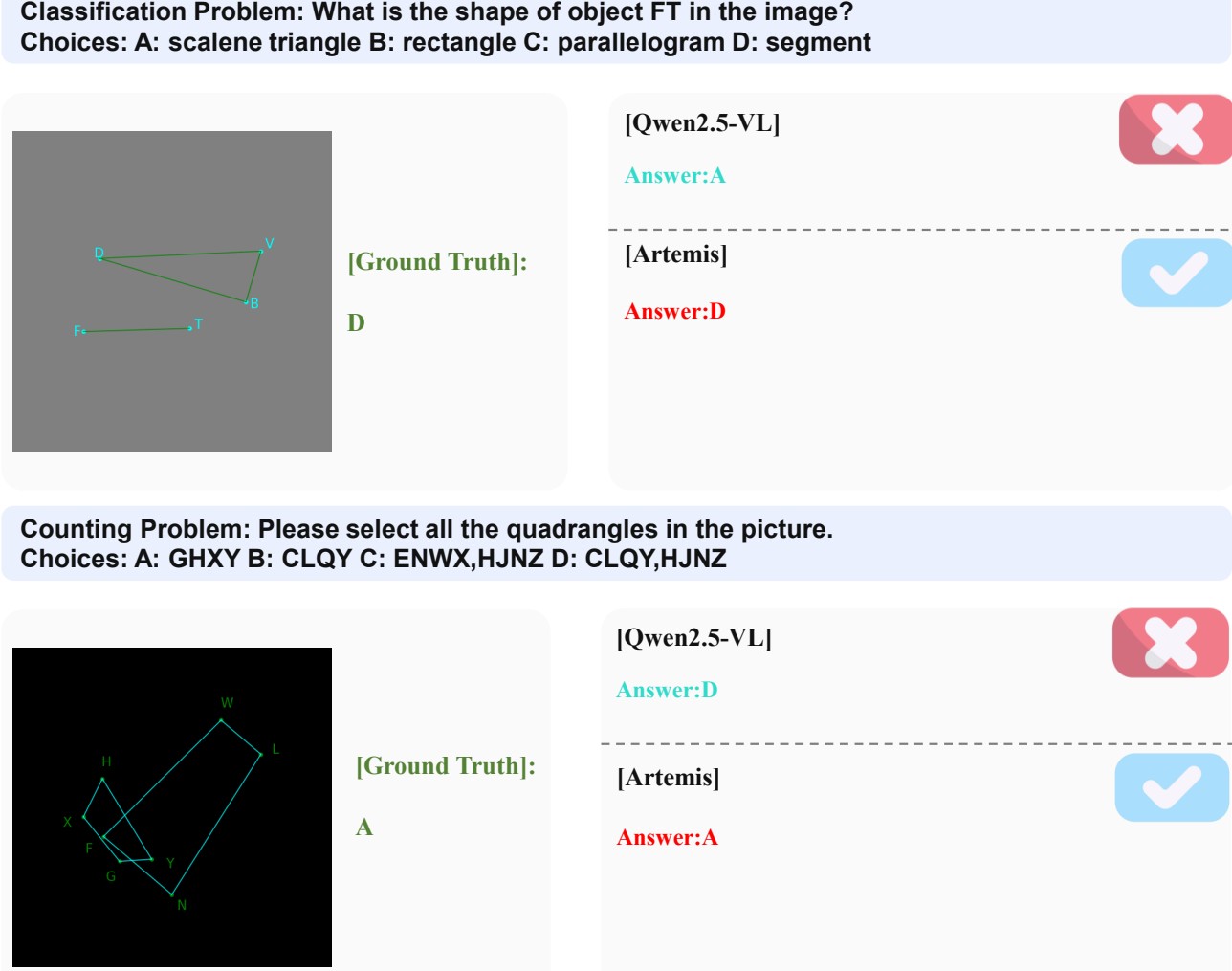

**Classification Problem: What is the shape of object FT in the image?**
**Choices: A: scalene triangle B: rectangle C: parallelogram D: segment**

[Qwen2.5-VL]

Answer:A

[Ground Truth]:

D

[Artemis]

Answer:D

**Counting Problem: Please select all the quadrangles in the picture.**
**Choices: A: GHXY B: CLQY C: ENWX,HJNZ D: CLQY,HJNZ**

[Qwen2.5-VL]

Answer:D

[Ground Truth]:

A

[Artemis]

Answer:A

*Figure 13.* Qualitative mathematical perception results of Artemis compared with Qwen2.5-VL on MATHGLANCE, showing examples from the mathematical shape Classification and mathematical shape Counting tasks. For brevity, full prompts are omitted. Green: ground truth; Cyan: Qwen2.5-VL; Red: Artemis answer.

**Section 4.2 (Main paper)**, although Artemis has not been trained on counting tasks or related data, it is able to internally **enumerate** instances during the <think> phase and **directly** produce accurate numeric counts in these cases. In contrast, Perception-R1 and Qwen2.5-VL-3B rely on post-processing, where counts are derived from detected objects. Such statistical counting does not reflect true learning of counting as a perceptual ability, but rather represents a simplified variant of detection-based perception.

For example, in Case 1 (Fig. 11), Perception-R1 over-detects one bounding box (box 7), while Qwen2.5-VL-3B outputs repeated boxes (boxes 2–5 and 6–9 are duplicates). In Case 2 (Fig. 12), both methods fail to detect some instances. Taken together with the quantitative results in the main paper, these visualizations suggest that Artemis develops enhanced perception capabilities. By leveraging structured visual reasoning, the model acquires perceptual skills that seamlessly transfer to counting tasks, resulting in counting behavior remarkably similar to **human intuition**.

### F.3. Additional Qualitative Results

Futhermore, we present visualizations of Artemis on the benchmarks of MATHGLANCE (Fig. 13 and Fig. 14), COCO Val 2017 (Fig. 15), and the LISA grounding test (Fig. 16). These cases provide several observations that complement the quantitative results reported in the main paper.

First, the perception capabilities learned from natural images can transfer to visually distinct domains, such as mathematical

**Grounding Problem: rectangle CDXM.**

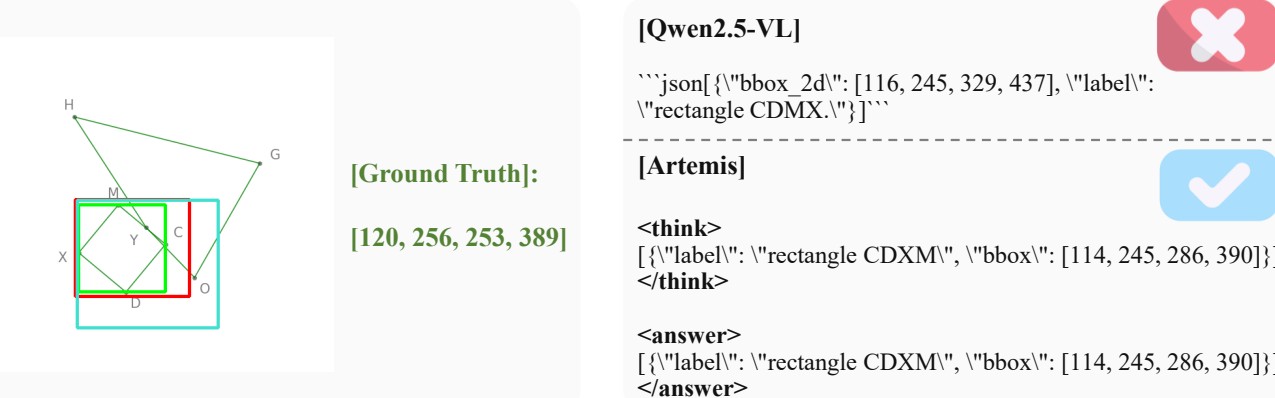

**[Qwen2.5-VL]** ❌

```json[{\"bbox_2d\": [116, 245, 329, 437], \"label\": \"rectangle CDMX.\"}]```

**[Ground Truth]:**

**[120, 256, 253, 389]**

**[Artemis]** ✔️

**\<think\>**
[{\"label\": \"rectangle CDXM\", \"bbox\": [114, 245, 286, 390]}]
**\</think\>**

**\<answer\>**
[{\"label\": \"rectangle CDXM\", \"bbox\": [114, 245, 286, 390]}]
**\</answer\>**

**Relation Problem: In the image, where is circle DGKR in relation to scalene triangle COY?**
**Choices: A: bottom right B: bottom left C: top left D: top right**

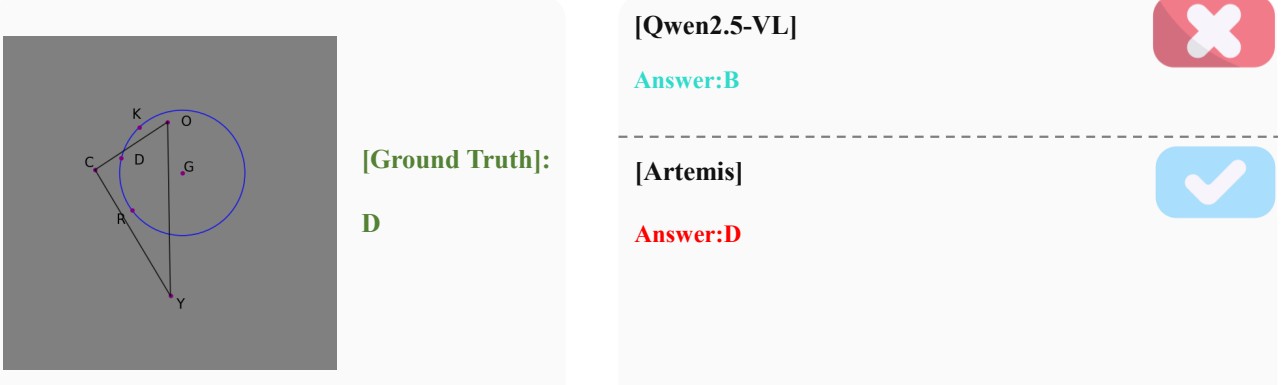

**[Qwen2.5-VL]** ❌

Answer:B

**[Ground Truth]:**

D

**[Artemis]** ✔️

Answer:D

*Figure 14.* Qualitative mathematical perception results of Artemis compared with Qwen2.5-VL on MATHGLANCE, showing examples from the mathematical Visual Grounding and mathematical Relation Identification tasks. For brevity, full prompts are omitted. Green: ground truth; Cyan: Qwen2.5-VL; Red: Artemis answer.

diagrams, improving performance on related perception tasks. For example, in the MATHGLANCE counting problem (Fig. 13), Artemis correctly identifies the only quadrilateral in the scene, whereas Qwen2.5-VL incorrectly selects option D, whose referenced vertices do not even form a valid shape in the diagram. In the MATHGLANCE relation problem (Fig. 14), Artemis also succeeds in selecting the correct relational configuration by accurately perceiving the relative positions of the relevant shapes, while Qwen2.5-VL again makes an incorrect choice.

Second, for detection and reasoning grounding, we observe an overall enhancement of scene-level perceptual capabilities. In detection cases (Fig. 15), Artemis does not miss any ground-truth objects, suggesting improved perception in natural scenes. For LISA grounding test, which requires knowledge-intensive reasoning, Artemis is able to locate relevant visual evidence to support its answers. For instance, in the third case of Fig. 16, the model first perceives the relevant visual evidence, identifying the person mentioned in the query as well as the ladder, and then uses this evidence to infer the final answer. The visualizations illustrate that structured visual reasoning training leads to stronger object-centric perception and more accurate reasoning-based grounding across diverse tasks.

## Detection Problem: List all visible objects in the image with their bounding boxes.

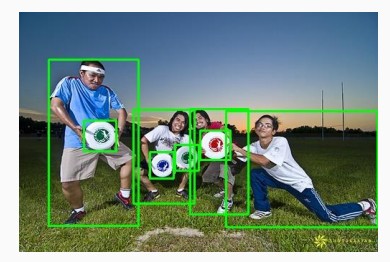 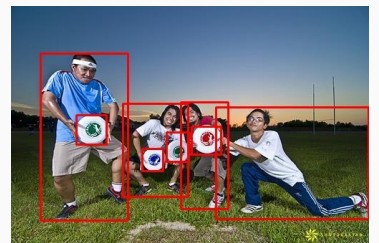

[Artemis]

<answer>
[{\"label\": \"person\", \"bbox\": [42, 67, 165, 300]},
{\"label\": \"person\", \"bbox\": [158, 137, 249, 267]},
{\"label\": \"person\", \"bbox\": [239, 135, 306, 284]},
{\"label\": \"person\", \"bbox\": [289, 142, 503, 298]},
{\"label\": \"frisbee\", \"bbox\": [92, 153, 135, 194]},
{\"label\": \"frisbee\", \"bbox\": [185, 200, 214, 231]},
{\"label\": \"frisbee\", \"bbox\": [219, 177, 250, 218]},
{\"label\": \"frisbee\", \"bbox\": [255, 169, 296, 208]}]
</answer>

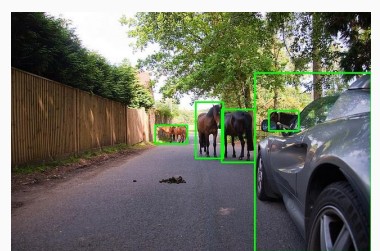 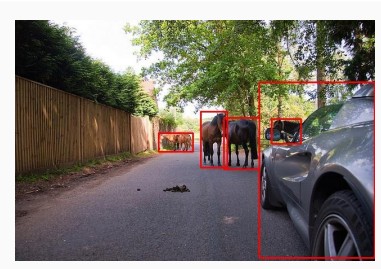

[Artemis]

<answer>
[{\"label\": \"horse\", \"bbox\": [333, 160, 380, 257]},
{\"label\": \"horse\", \"bbox\": [374, 171, 437, 260]},
{\"label\": \"horse\", \"bbox\": [257, 197, 318, 230]},
{\"label\": \"horse\", \"bbox\": [459, 173, 512, 216]},
{\"label\": \"car\", \"bbox\": [437, 109, 643, 415]}]
</answer>

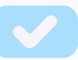

*Figure 15.* Qualitative object detection results of Artemis on COCO Val 2017. Green: ground truth; Red: Artemis answer;

**Groudning Problem:**
**the container that is being held by a person and is about to pour liquid**

Solution: [557, 247, 669, 649]

**Grounding Problem:**
**What objects are commonly placed on a table to provide additional illumination to a room?**

Solution: [196, 25, 319, 207]

**Grounding Problem:**
**It can be difficult for people to climb up a bare pole and inspect or repair the upper part. What object is the person in the picture relying on to accomplish this task?**

Solution: [417, 165, 603, 1119]

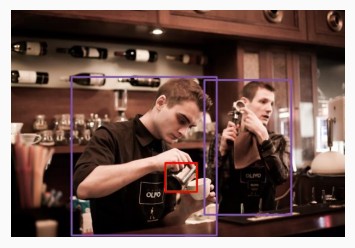 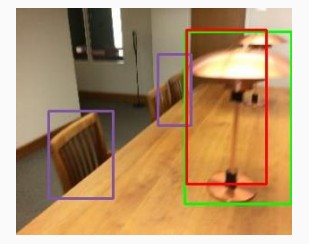 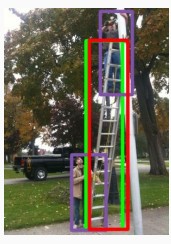

[Artemis]

<think>
[{\"label\": \"the container that is being held by a person and is about to pour liquid\", \"bbox\": [556, 547, 670, 650]},
{\"label\": \"person\", \"bbox\": [218, 239, 741, 806]},
{\"label\": \"person\", \"bbox\": [697, 250, 1010, 732]}]
</think>

<answer>
[{\"label\": \"the container that is being held by a person and is about to pour liquid\", \"bbox\": [556, 547, 670, 650]}]
</answer>

[Artemis]

<think>
[{\"label\": \"objects are commonly placed on a table to provide additional illumination to a room\", \"bbox\": [190, 24, 279, 195]},
{\"label\": \"chair\", \"bbox\": [36, 114, 108, 211]},
{\"label\": \"chair\", \"bbox\": [158, 51, 195, 130]}]
</think>

<answer>
[{\"label\": \"objects are commonly placed on a table to provide additional illumination to a room\", \"bbox\": [190, 24, 279, 195]}]
</answer>

[Artemis]

<think>
[{\"label\": \"person\", \"bbox\": [490, 17, 653, 438]},
{\"label\": \"person\", \"bbox\": [344, 742, 519, 1124]},
{\"label\": \"ladder\", \"bbox\": [434, 163, 630, 1119]}]
</think>

<answer>
[{\"label\": \"ladder\", \"bbox\": [434, 163, 630, 1119]}]
</answer>

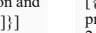 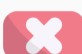

*Figure 16.* Qualitative reasoning-based grounding results of Artemis on the LISA grounding test set. Green: ground truth; Purple: Artemis reasoning boxes; Red: Artemis answer.

