# OpenReview forum: "Artemis: Structured Visual Reasoning for Perception Policy Learning"
_ICML.cc/2026/Conference — ICML 2026 regular_

### Official Review · Reviewer_JgMA · 2026-02-23

**Soundness:** 4
**Presentation:** 4
**Significance:** 2
**Originality:** 1
**Overall Recommendation:** 4
**Confidence:** 4

**Summary:**

Artemis introduces a type of chain of thought reasoning for visual reasoning/recognition tasks which instead of reasoning purely in language, reasons using grounded bounding boxes in the image. The paper introduces a CoT reasoning technique and RL training regime and fine-tunes a reasoning model to improve performance on visual recognition and reasoning tasks. Results show fairly consistent performance increases across a variety of tasks.

**Compliance With Llm Reviewing Policy:**

Affirmed.

**Final Justification:**

See my response to authors, but my major concerns were all addressed, so I am comfortable recommending the paper.

**Key Questions For Authors:**

See weaknesses for more details:
1. questions about comparison to prior work, especially Sarch et. al. and
2. experimental comparisons to Sarch.
3. Why Perception-R1 outperforms on COCO despite this method being trained on it

**Limitations:**

This could be improved. How would human-parsible chain of thought for visual models affect societal outcomes. I think authors could put a bit more thought into this.

**Strengths And Weaknesses:**

Strengths
- The primary idea: doing chain of thought through grounded bounding boxes in the image is intuitive and well thought out.
- The paper is wellwritten, easy to follow and makes good use of visuals.
- There are a large number of results in a variety of different settings and Artemis mostly seems to outperform baselines (however, see weaknesses for caveats to this)
- Qualitative examples from the CoT reasoning are very compelling. This is a distinct advantage even beyond topline accuracy as it gives human parsible evidence

Weaknesses
- I think that this paper might be missing an important prior work: Grounded Reinforcement Learning for Visual Reasoning, Sarch et al. Neurips 2025. As far as I can tell it proposes a very similar visual chain of thought reasoning method, using a variation of GRPO, including on perception tasks. I think this is quite similar and my preliminary assessment is that the works are quite similar, but would appreciate hearing from authors on this as well as seeing experimental comparisons between the methods. While the other cited works related to visual reasoning are said to be targeting different domains (e.g. Huang et al., 2025; Chen et al., 2025, Xiong et al., ..) the related work does not really explain why the methods of these papers are different and not applicable to the target domain of this paper. Indeed authors compare to several of these including VLM-R1 and the results seem improved, but not overwhelmingly so.
- In some of the comparisons, it's not clear from the text what the backbones of all of these are, and at least Perception-R1 definitely uses a Qwen2.0 backbone. Authors should clearly state the backbones of each of the methods in the result tables and discuss the implecations of comparing baselines with different backends.
- Results are mostly convincing. There is some improvements on many of these tasks, but then for the COCO experiments, it seems to be outperformed by Perception-R1. This is especially odd since Artemis is specifically trained on COCO data, which you would think would give it more of an advantage.

---

> ### Author Rebuttal · Authors · 2026-03-31
>
> **Q1: Comparison to prior work ViGoRL.**
>
> A1: We thank the reviewer for pointing out ViGoRL. The overlap is only at a high level: both question whether purely linguistic reasoning is sufficient for perception. However, the methods differ fundamentally in the supervision and training paradigm.
>
> - ViGoRL anchors intermediate steps to point coordinates mainly to enable revisiting local regions, and relies primarily on answer-level supervision within a task-specific setup, without explicit supervision on intermediate perceptual states.
> - In contrast, Artemis represents reasoning as structured visual evidence (label, bounding box) and applies explicit object-centric supervision on these intermediate states, learning a unified perception policy across tasks.
>
> In short, ViGoRL explores *where to look*, whereas Artemis learns *what constitutes correct intermediate visual evidence*. This difference is also reflected empirically:
>
> | Model        | COCO mAP | PixMo | M.G. |
> |--------------|----------|-------|------|
> | ViGoRL-3B  | not support | 26.8 | 3.6  |
> | **Artemis-3B** | 31.0     | 81.4  | 49.3 |
>
> Since ViGoRL predicts points rather than boxes, we further align evaluation by converting our boxes to center points. We report Point-in-BBox (PIB), i.e., whether the predicted point falls inside the GT box, and Mean Dist (M.D.), i.e., the pixel distance to the GT box center:
>
> | Model        | RefCOCOg val (PIB↑ / M.D.↓) | LISA test (PIB↑ / M.D.↓) |
> |--------------|----------------------------|------------------------|
> | ViGoRL-3B    | 63.6 / 109.5             | 10.6 / 479.9         |
> | **Artemis-3B**   | 92.1 / 26.1              | 61.8 / 123.1         |
>
> Even under aligned evaluation, the gap remains large, indicating that coordinate-based revisit reasoning without intermediate supervision is insufficient, while structured and verifiable intermediate supervision leads to substantially better performance.
>
> ----
>
> **Q2: Differences from prior visual reasoning works.**
>
> A2: We thank the reviewer for the suggestion and will clarify this in the revision. Prior works differ along two aspects:
>
> - Some focus on task-specific domains (e.g., Chen et al. for math reasoning, Xiong et al. and Yu et al. for document understanding).
> - Others (e.g., Huang et al.) study visual perception under a perception-policy setting.
>
> However, across these works, reasoning is typically expressed in a free-form manner, and supervision is applied to final answers rather than intermediate reasoning states. In contrast, Artemis represents reasoning as structured spatial (label, bbox) evidence and applies supervision directly to these predictions, leading to more reliable and consistent performance across visual perception tasks.
>
> ----
>
> **Q3:  The results seem improved, but not overwhelmingly so.**
>
> A3: We respectfully disagree with the assessment that the improvements are not substantial.
>
> - Artemis shows clear absolute gains over VLM-R1 across multiple benchmarks:
>   - COCO val mAP: +9.4
>   - MATHGLANCE: +14.9
>   - MMBench: +8.6
>   - POPE: +9.3
> - We also observe consistent gains over other RL-based methods, e.g., +3.3 on PixMo-val over Perception-R1.
>
> These improvements are consistent across both perception-heavy benchmarks and general MLLM benchmarks, as shown in Tables 1–4.
>
> ----
>
> **Q4: Why Perception-R1 outperforms on COCO.**
>
> A4: While Perception-R1 achieves slightly higher COCO mAP, it is trained as a task-specific model using COCO alone, optimizing for in-domain performance. In contrast, Artemis is designed as a unified perception policy that balances multiple tasks and prioritizes generalization.
>
> - Artemis still achieves a competitive 31.0 mAP on COCO.
> - More importantly, a single unified model demonstrates strong performance across diverse tasks, including grounding, counting, structured perception, and general MLLM benchmarks (Tables 1, 3, 4, 5).
>
> This reflects a trade-off: Perception-R1 optimizes for a single task, whereas Artemis maintains strong performance while generalizing across domains.
>
> ----
>
> **Q5: Backbone differences and comparison fairness.**
>
> A5: We thank the reviewer for the suggestion and will explicitly list all backbones in the revised tables. We agree that different backbones can introduce variations. Our comparisons aim to provide a broad reference across representative methods, rather than a strictly backbone-matched setting. Importantly:
>
> - All ablations are conducted with the same backbone (Qwen2.5-VL-3B), isolating the effect of our reasoning framework.
> - Artemis (3B) outperforms several methods using larger 7B backbones (e.g., Vision-R1, VisionReasoner) on multiple benchmarks.
>
> This suggests that the gains are not solely due to backbone scale, but come from the proposed reasoning design.
>
> ----
>
> **Q6: Societal impact of human-parsable reasoning.**
>
> A6: Human-parsable reasoning helps users inspect and verify model decisions, especially in perception tasks, rather than relying on ambiguous free-form explanations.

---

> > ### Author Rebuttal · Reviewer_JgMA · 2026-04-02
> >
> > My concerns are fully addressed. I think there may have been a slight miscommunication; I did agree with authors that their results seemed substantial with the exception of the COCO experiment, which I was curious about. I think the author's answer on this cleared up my question.
> > Authors also addressed my concerns with the comparisons to this and other methods here and in response to another reviewer. Contingent on these results and discussion about these prior works is included in the final paper, I think this addresses all my concerns and I'll be adjusting my score accordingly.

---

> > > ### Author Response · Authors · 2026-04-04
> > >
> > > We sincerely thank the reviewer for the positive and helpful feedback, and are glad the concerns are addressed. We also greatly appreciate your adjustment of the score. We will include these results and the discussion about these prior works in the final paper, and we will incorporate the results and discussion in the final revision.

---

### Official Review · Reviewer_5dNc · 2026-03-10

**Soundness:** 3
**Presentation:** 3
**Significance:** 3
**Originality:** 2
**Overall Recommendation:** 5
**Confidence:** 5

**Summary:**

This paprr introduces Artemis, a GRPO-based post-training framework for Qwen2.5-VL-3B that replaces free-form linguistic reasoning with structured visual reasoning, where intermediate thoughts are represented as (label, bounding-box) pairs. The model is trained on a roughly 77k COCO/RefCOCO-derived post-training set with an additional 80k COCO detection cold-start, and is evaluated on visual grounding, object detection, counting, reasoning grounding, mathematical diagram perception, and broader multimodal benchmarks. Empirically, the paper shows strong gains on perception-centric tasks and meaningful transfer to several out-of-domain settings.

**Compliance With Llm Reviewing Policy:**

Affirmed.

**Final Justification:**

See my response to the authors, but my major concerns were all addressed, so I am comfortable recommending the paper.

**Key Questions For Authors:**

- How much of the gain comes from stronger perceptual supervision, versus the structured process format itself? Since detection training omits <think> and the method also uses a substantial detection cold-start, I would like a cleaner disentanglement of supervision source, reward design, and reasoning format.

- Is the linguistic baseline unfairly weak? The current “Ling. Rsn.” comparison appears to use linguistic reasoning with only final-answer supervision, rather than process-supervised natural language or an interleaved language+grounding format.

- What are the reasoning lengthand latencies for None, Ling. Rsn., and S.V. Rsn., and what happens under equal token budgets? A length-controlled comparison would directly test the alternative explanation that Artemis works partly by keeping the reasoning trace short, local, and visually anchored.

- How well does Artemis handle revisit / longer-horizon visual reasoning? I would like either a comparison to recent revisit/look-back style evaluations (like V* Bench, HR-Bench 4K), or a direct analysis of how attention to visual input evolves over the course of reasoning. Also maybe discuss with coparission to works like Qwen Look Again (https://arxiv.org/abs/2505.23558), or Look-Back (https://arxiv.org/abs/2507.03019v1) That triy to tackle the same goudnign idea buut ina different way.

**Strengths And Weaknesses:**

The empirical recipe is strong and mostly convincing. The paper evaluates across a broad set of perception-heavy benchmarks, and the gains over the Qwen2.5-VL-3B base model are large on the tasks that most directly matter for the paper’s thesis, including grounding, detection, counting, LISA, and MATHGLANCE. For example, COCO detection improves from 15.4 to 31.0 mAP, RefCOCOg valAvg improves from 63.9 to 67.1, and MATHGLANCE Avg improves from 33.1 to 49.3. But this is where i think the papermore strongly establishes a recipe than a mechanism.  The training setup changes several things at once: grounding rewards, detection rewards, structured visual reasoning rewards, a substantial 80k detection cold-start, and a detection task that explicitly omits <think> and directly supervises final box outputs.

The paper is generally clear, well structured, and easy to follow. The motivation is intuitive, the reward design is readable, and the ablation section is useful. My main presentation concern is not clarity of writing, but a little overclaiming. The paper often frames the result as evidence for a principled route toward “general perception policies,” yet the strongest evidence is concentrated on perception-centric tasks, while the broader multimodal benchmark gains are comparatively modest (for example, MMBench 79.1->79.3, MMVet 60.0->61.4, MMStar 53.8->55.9, with AI2D essentially flat and BLINK slightly lower)

I view the paper as significant because perception-focused RL post-training is an important and still unresolved problem, and this work presents a simple, clean, and practically useful harness. Even if the deeper mechanism claim is not yet fully proven, the empirical finding that spatially aligned, verifiable supervision transfers well across grounding, detection, counting, and diagram perception is valuable and likely to influence follow-up work.

I think the novelty is stronger at the level of harness design than at the level of a new understanding of visual reasoning. The main reason is that the paper does not yet cleanly disentangle whether the gains come from the reasoning format itself, from stronger perceptual supervision, or from shorter and more visually anchored traces that avoid some of the failure modes of long linguistic CoTs.

---

> ### Author Rebuttal · Authors · 2026-03-31
>
> **Q1: Supervision vs. reasoning format.**
>
> A1: We agree that a cleaner disentanglement is important. To isolate the contributions of supervision source, cold-start, and reasoning format, we augmented Table 5.
>
> | Grd | Det | S.V. Rsn | Cold | RefCOCOg val | COCO mAP | M.G. |
> |-----|-----|-----------|------|---------------|----------|------|
> | ✓   | ✗   | ✗         | ✓    | 86.6          | 27.5     | 43.1 |
> | ✗   | ✓   | ✗         | ✓    | 51.6          | 30.6     | 44.0 |
> | ✓   | ✓   | ✗         | ✓    | 87.1          | 30.7     | 43.7 |
> | ✓   | ✓   | ✓         | ✓    | 87.3          | 31.0     | 49.3 |
> | ✓   | ✓   | ✓         | ✗    | 87.2          | 29.8     | 46.1 |
>
> - Combining grounding and detection supervision alone (Rows 1–3) does not yield meaningful OOD gains, with M.G. remaining around 43.7.
> - Under the exact same supervision setup, adding S.V. Rsn (Row 3 → 4) brings a clear OOD improvement (43.7 → 49.3), while also improving RefCOCOg and COCO mAP.
> - Removing the cold-start (Row 4 → 5) causes a moderate drop, but the model still remains above the no-S.V. Rsn setting on M.G. (46.1 vs. 43.7), indicating that cold-start helps initialization but is not the main source of the gain.
>
> Overall, these results more cleanly disentangle the factors: stronger supervision is helpful, cold-start improves initialization, but the structured reasoning format is the main driver of the generalization gain.
>
> **Q2: Fairness of the linguistic baseline.**
>
> A2: We agree that this comparison should be clarified. Process supervision for free-form or interleaved language+grounding reasoning is inherently difficult to realize with deterministic rules, and in practice usually relies on an LLM judge to evaluate intermediate steps, which introduces additional noise. To ensure a fair comparison and isolate the effect of the representation format, we therefore compare our structured model under the same final-answer-only supervision as the “Ling. Rsn.” baseline. Under this matched setting, simply adopting the structured format already improves performance: (COCO mAP: 30.4 → 30.7, RefCOCOg: 86.4 → 87.1).
>
> This shows that the gain is not only due to stronger supervision in this comparison. Furthermore, because our [label, bbox] format naturally disentangles labels and coordinates, it also enables precise rule-based process supervision, which leads to the additional gains in the final model (COCO mAP: 31.0; RefCOCOg: 87.3).
>
> **Q3: Reasoning length and latency.**
>
> A3: Thank you. Our empirical results on RefCOCOg test show the opposite of the reviewer’s hypothesis: S.V. Rsn. actually produces longer reasoning traces than the linguistic baseline.
>
> | Method    | Mean Tokens | Mean Latency (s) |
> |-----------|-------------|------------------|
> | None      | 108.7       | 1.68             |
> | Ling. Rsn.| 194.7       | 2.91             |
> | S.V. Rsn. | 270.4       | 4.03             |
>
> - All methods are already evaluated under the same max token length (1024).
> - For autoregressive models, a meaningful length-controlled comparison requires a budget that is long enough to cover the typical outputs of all methods.
> - Otherwise, using a much smaller tokens (e.g., 128) would disproportionately truncate methods that naturally require longer reasoning traces, and the comparison would mainly reflect truncation effects rather than reasoning quality.
>
> Therefore, our gains cannot be explained by keeping the reasoning trace short. Instead, the results show that Artemis achieves stronger performance despite using longer structured reasoning traces.
>
> **Q4: How well does Artemis perform on revisit-style benchmarks, and how does it compare with methods such as Look-back or Qwen Look Again?**
>
> A4: We evaluate Artemis on V* and HR-Bench4K and observe consistent gains over the Qwen2.5-VL-3B:
>
> - V*: 75.4 → 76.4
> - HR-Bench4K: 66.5 → 67.6
>
> | Method         | V* (Overall / Time (h)) | HR-Bench4K (Overall / Time (h)) | RefCOCOg | Pixmo | POPE |
> |----------------|--------------------------|----------------------------------|----------|-------|------|
> | Qwen2.5-VL-7B  | 77.0 / 0.2               | 68.3 / 4.0                       | 87.2     | 71.6  | 86.4 |
> | Look-back-7B   | 78.0 / 2.7               | 69.2 / 108.0                     | 70.7     | 71.4  | 86.3 |
> | Qwen2.5-VL-3B  | 75.4 / 0.1               | 66.5 / 0.9                       | 85.1     | 58.0  | 85.9 |
> | **Artemis-3B** | 76.4 / 0.1               | 67.6 / 3.8                       | 87.3     | 81.4  | 88.6 |
>
> While Look-back achieves slightly higher absolute scores, its relative gains over its Qwen2.5-VL-7B are comparable to ours (~+1.0), but at much higher cost (e.g., 108h vs. 4h). Moreover, its improvements do not transfer to general perception benchmarks, where it underperforms its backbone on RefCOCOg, Pixmo, and POPE. For Qwen Look Again, no public code or checkpoints are available. Overall, Artemis achieves comparable gains with much better efficiency and more consistent cross-task generalization.

---

> > ### Author Rebuttal · Reviewer_5dNc · 2026-04-02
> >
> > Thank you, this addresses my concerns. I'll adjust my score accordingly.

---

> > > ### Author Response · Authors · 2026-04-04
> > >
> > > We sincerely thank the reviewer for the positive feedback and for recognizing that the concerns have been addressed. We greatly appreciate your thoughtful evaluation and your willingness to adjust the score. Your feedback has been very valuable in improving the clarity and presentation of our work, and we will incorporate the results and discussion in the final revision.

---

### Official Review · Reviewer_HSCy · 2026-03-11

**Soundness:** 2
**Presentation:** 3
**Significance:** 2
**Originality:** 2
**Overall Recommendation:** 3
**Confidence:** 4

**Summary:**

The authors propose Artemis, which uses GRPO + a novel reward formulation to encourage the model to think in bounding boxes, rewarding informative boxes in the reasoning chain. The authors show improvements over perception-R1, which does GRPO without thinking due to its harm on performance.

**Compliance With Llm Reviewing Policy:**

Affirmed.

**Final Justification:**

The rebuttal partially addressed my concerns but some concerns remain:

They claim zero shot improvements based on their structured reasoning, but based on their responses, it remains unclear how much of the zero-shot improvements come from the thinking traces. If their central claim is "perception policies benefit most not from eliminating thinking, but from structuring it spatially", then all evaluations should use the structured thinking, and it would be misleading to claim their thinking generalizes zero shot without using it at test time. Either the claim should be revised or they should show evidence that the thinking is used and helping on all the evaluations. Other things like some of their reported baseline numbers (mmvet) seem lower the official report also raises concerns.

However, for grounding and counting tasks, their training does show improvements, so on this basis I do see the work as valuable.

**Key Questions For Authors:**

- How does the method performance compare to [1,2,3,4]? How is the work different from these methods?
- What annotation overhead does the dense annotation add over other methods? How will this scale to larger data without dense annotations?

**Limitations:**

The authors could include more discussion on the limitations of their work

**Strengths And Weaknesses:**

Strength
- Tackles an important problem of how to perform reasoning in perception-heavy tasks
- Designs a new GRPO reward for grounded reasoning

Weaknesses
- The authors are missing comparisons and discussions to highly relevant methods that do grounded reasoning [1,2,3,4].
- The method requires dense annotations of the objects in the scene.
- The overly structured reasoning may over-restrict reasoning behaviors. Why not explore more natural interleaved text and grounding?

[1] Pixel Reasoner: Incentivizing Pixel-Space Reasoning with Curiosity-Driven Reinforcement Learning
[2] Grounded Reinforcement Learning for Visual Reasoning
[3] DeepEyes: Incentivizing "Thinking with Images" via Reinforcement Learning
[4] Molmo and PixMo: Open Weights and Open Data for State-of-the-Art Vision-Language Models

---

> ### Author Rebuttal · Authors · 2026-03-31
>
> **Q1: The authors are missing comparisons and discussions to highly relevant methods that do grounded reasoning [1,2,3,4].**
>
> A1: We thank the reviewer for the suggestion and will include a detailed comparison in the revised manuscript.
>
> | Method    | Paradigm | Re-enc | Prog Rwd | Summary |
> |-----------|----------|--------|----------|---------|
> | [1][2][3] | RL       | ✓      | ✗        | Multi-turn image manipulation (e.g., crop, zoom) with additional visual selections and visual re-processing for thinking; no intermediate supervision. |
> | [4]       | SFT      | ✗      | ✗        | Large-scale perception-aligned dataset construction for MLLM SFT; supporting point-based outputs; no intermediate supervision. |
> | **Artemis**   | RL       | ✗      | ✓        | Single-turn thinking with structured format (label-box); supports intermediate spatial perception supervision. |
>
> - Methods [1, 2, 3] (Pixel Reasoner, ViGoRL, DeepEyes) mainly focus on multi-turn interaction, improving performance by iteratively modifying the visual input through zooming, cropping, or grounded revisit (Re-enc in the table).
> - However, these approaches rely on answer-level rewards, without supervision on intermediate perceptual states (Prog Rwd in the table), and depend on additional visual selections and re-processing during reasoning.
>
> In contrast, Artemis performs single-turn structured reasoning under a fixed visual input.
>
> - By representing reasoning as explicit (label, box) pairs rather than image revisits, Artemis enables verifiable structured visual reasoning with intermediate spatial supervision, providing direct supervision on the reasoning process itself.
> - Compared to [4] (Molmo/PixMo), which scales perception through large-scale SFT on high-quality data, Artemis focuses on policy learning through RL.
> - Our structured visual reasoning reward enables the model to learn a robust perception policy that generalizes across diverse domains.
>
> As shown in our results, even with a smaller 3B backbone, Artemis significantly outperforms [1, 2, 3] and demonstrates superior generalization over [4] on out-of-domain tasks.
>
> |Model|RefCOCOg val|COCO mAP|MATHGLANCE|PixMo val|
> |---|---|---|---|---|
> |Vision Reasoner-7B|63.1|12.8|59.2|71.8|
> |ViGoRL-3B|-|-|3.6|26.8|
> |DeepEyes-7B|86.7|7.4|30.6|45.0|
> |Molmo-7B|-|-|46.6|-|
> |**Artemis-3B**|87.3|31.0|49.3|81.4|
>
> ----
> **Q2:The method requires dense annotations of the objects in the scene.
> What annotation overhead does the dense annotation add over other methods? How will this scale to larger data without dense annotations?**
>
> A2: We clarify that our method introduces zero additional manual annotation overhead.
>
> - All supervision signals are simply structurally reorganized from existing public datasets (e.g., COCO, LLaVA-665k, Vision-R1). Please kindly refer to Appendix "C. Data Construction" for more details.
>
> Regarding scalability of Artemis, training data can be scaled without human labeling costs:
>
> - Potential direction for scaling without human labeling using standard automated data-construction pipelines: extend unannotated image-level dataset to object-level structured supervision by pairing off-the-shelf detectors (e.g., DINOX) for bounding box generation with MLLMs for detailed region descriptions.
>
> Thus, scaling our approach may require only automated synthesis, not additional human effort.
>
> ----
>
> **Q3: Why not explore more natural interleaved text and grounding?**
>
> A3: Thank you. The main reason is that natural language is inherently ambiguous and fails to process supervision, harming perceptual tasks (Artemis's key motivation). To address this concern, we compare Artemis with ViGoRL (interleaved text and grounding thinking chains); Artemis achieves +45.7 on MATHGLANCE and +54.6 on PixMo-val; for RefCOCOg/LISA-test (+28.5/+51.2), please kindly refer to **Rev. JgMA-Q1**.
>
> More explanations:
>
>  (i) why natural language difficult to explicitly supervise and verify:
> - In such free-form formats, it is often unclear which textual phrase corresponds to which predicted bounding box.
> - Consequently, it is difficult to define reliable and deterministic rule-based evaluation over the reasoning process.
>
> (ii) by enforcing a strict [label, bbox] structured format, we eliminate this ambiguity:
>
> - This design allows rewards to be computed in a fully rule-based and deterministic manner over structured reasoning outputs, ensuring precise, high-quality supervision without the ambiguity of unstructured text.
>
> Overall, this design choice directly reflects the core motivation of our work.
>
> ----
>
> **Q4: The authors should discuss limitations.**
>
> A4: Our current structured visual reasoning is limited to simple label–box representations and static images.
>
> - Extending it to richer representations and video is left for future work.

---

> > ### Author Rebuttal · Reviewer_HSCy · 2026-04-02
> >
> > My concerns are partially addressed but I still have the following concerns:
> > - What does the grounded reasoning look like for MATHGLANCE? Figure 12 shows no reasoning is given. Are the baselines also evaluated without reasoning? I have the same question for the benchmarks in Table 5. If not trained for these tasks, does the model reasoning boxes still look reasonable?
> > - I understand that counting and math are not trained in Artemis, but what tasks do the other works [1,2,3,4] train and would we expect Artemis to outperform them with broad-task training? The rebuttal is helpful but I still have some concerns about this structured reasoning training being more specific to grounding and counting types of tasks. For example, obtaining reliable annotations for non-object areas ("the area in between two objects", "the edge of the road") are difficult to supervise.

---

> > > ### Author Response · Authors · 2026-04-04
> > >
> > > We thank the reviewer's response and apologize for any lack of clarity in our original presentation; we provide a detailed clarification below.
> > >
> > > ---
> > >
> > > **Q1**:  What does the grounded reasoning look like for MATHGLANCE? Figure 12 shows no reasoning is given. Are the baselines also evaluated without reasoning? I have the same question for the benchmarks in Table 5. If not trained for these tasks, does the model reasoning boxes still look reasonable?
> > >
> > > **A1:** Thank you for your review and thoughtful discussion.
> > >
> > > We believe the referenced “Table 5” for benchmark results should be Table 4 (Table 5 presents our modular ablations).
> > >
> > > **During MATHGLANCE and general MLLM benchmark inference, we do not explicitly control model responses (i.e., whether to produce reasoning traces or directly answer).** We keep the same evaluation pipeline as the baseline models and use the default prompts for both MATHGLANCE and the general MLLM benchmarks (Table 4), ensuring a fair comparison. We have explicitly detailed this in the manuscript (Sec. 4.2) and the supplementary material (Sec. D). We observe that our model may autonomously trigger an explicit reasoning process when encountering grounding-related tasks (e.g., the mathematical visual grounding example in Figure 14 of MATHGLANCE); for other sub-tasks in MATHGLANCE or general MLLM benchmarks, such as POPE and MMVet, the model performs direct answering. **Whether the baselines trigger thinking or not depends on the model itself---how it follows the testing prompts and understands the tasks.**
> > >
> > > Overall, we maintain the identical evaluation protocol across all benchmarks for both our model and the baselines. Whether thinking is activated mainly depends entirely on the models' own behavior, which we cannot manually control. **Our work focuses on how to improve model's own perception ability and in what way to think is reasonable for perception  tasks**.
> > >
> > > We thus apply structural reasoning (box, label) thoughts due to their verifiable nature, and design rule-based intermediate supervision during RL training. The comprehensive experiments confirm that our model achieves stronger perception  ability and generalizes to broader perceptual tasks. Even without any task-specific designs or fine-tuning, our model achieves competitive zero-shot performances --- justifying our claim in Lines 061–064: a general perception system should operate consistently across diverse perceptual tasks and heterogeneous visual domains.
> > >
> > > ---
> > >
> > > **Q2**: I understand that counting and math are not trained in Artemis, but what tasks do the other works [1,2,3,4] train and would we expect Artemis to outperform them with broad-task training? The rebuttal is helpful but I still have some concerns about this structured reasoning training being more specific to grounding and counting types of tasks. For example, obtaining reliable annotations for non-object areas ("the area in between two objects", "the edge of the road") are difficult to supervise.
> > >
> > > **A2:** We understand the reviewer’s concern regarding the scalability of our approach. Methods [1–4] rely on broader but largely task-specific training mixtures (e.g., visual search in Pixel Reasoner, GUI/web grounding in ViGoRL, and chart/diagram reasoning in DeepEyes), and their gains are therefore mostly concentrated on domains seen during training. In contrast, Artemis is post-trained only on grounding and detection, **yet still demonstrates strong and consistent zero-shot transfer to out-of-domain benchmarks**. Our goal is precisely to test whether improving the model’s underlying perception and spatial reasoning ability can generalize beyond task-specific designs.
> > >
> > > The specificity concern: we would like to clarify that Artemis builds on a large-scale pre-trained base model and performs post-training, so it can leverage the semantic/world knowledge that the pre-trained model has already learned. Our contribution is to show that adding structured supervision on object-level visual entities is already sufficient to enhance perception ability of the model and enable transfer beyond the training tasks.
> > >
> > > We also note that examples such as “the area in between two objects” or “the edge of the road” are not standard object categories, but more abstract relation-defined or boundary regions. We agree that extending supervision to such regions could further strengthen the model. **However, reliably defining and supervising them at scale remains an open problem and is beyond the scope of the current work**. Importantly, such regions do not necessarily require direct annotation as standalone targets; they can often be inferred from object-level anchors and surrounding spatial context. **This may indicate that object-level structured supervision provides a scalable and practical foundation, without requiring exhaustive annotation of all abstract regions.**

---

### Decision · Program_Chairs · 2026-04-30

**Decision:**

Accept (regular)

**Comment:**

The paper describes a method for perception policy learning that replaces unstructured linguistic with structured visual CoT trained via GRPO and explicit supervision on intermediate spatial states.

On the positive side, the reviewers found the core idea intuitive and well-motivated; the writing was deemed clear, and most importantly empirical gains across grounding, detection, counting, and mathematical diagram perception benchmarks were seen as consistent.

On the negative side, some concerns were raised about conceptual overlap with prior grounded-reasoning work. On the experimental side, there were also concerns about insufficient empirical dissociation between supervision and reasoning, and the gains on general multimodal benchmarks were rated as modest. The rebuttal added experiments which satisfied two reviewers, both recommending accept and one adjusting their score upward.

One reviewer maintained weak reject citing remaining concerns about whether the zero-shot gains actually come from the thinking traces and about scope beyond grounding/counting. The authors engaged in a second-round of responses to clarify the evaluation protocol and scope but this reviewer did not engage further. The AC feels that the main concerns were generally well addressed, and in the absence of further engagement from the reviewer it is assumed that the concern is addressed.

Overall, the AC thus recommends the paper to be accepted.